# On the Bayes Inconsistency of Disagreement Discrepancy Surrogates

**Neil G. Marchant, Andrew C. Cullen, Feng Liu & Sarah M. Erfani**
School of Computing & Information Systems, University of Melbourne, Australia
{nmarchant,andrew.cullen,feng.liu1,sarah.erfani}@unimelb.edu.au

## Abstract

Deep neural networks often fail when deployed in real-world contexts due to distribution shift, a critical barrier to building safe and reliable systems. An emerging approach to address this problem relies on *disagreement discrepancy*—a measure of how the disagreement between two models changes under a shifting distribution. The process of maximizing this measure has seen applications in bounding error under shifts, testing for harmful shifts, and training more robust models. However, this optimization involves the non-differentiable zero-one loss, necessitating the use of practical surrogate losses. We prove that existing surrogates for disagreement discrepancy are not Bayes consistent, revealing a fundamental flaw: maximizing these surrogates can fail to maximize the true disagreement discrepancy. To address this, we introduce new theoretical results providing both upper and lower bounds on the optimality gap for such surrogates. Guided by this theory, we propose a novel disagreement loss that, when paired with cross-entropy, yields a provably consistent surrogate for disagreement discrepancy. Empirical evaluations across diverse benchmarks demonstrate that our method provides more accurate and robust estimates of disagreement discrepancy than existing approaches, particularly under challenging adversarial conditions.

## 1 Introduction

The reliability of deep neural networks is frequently undermined by distribution shift, where a model's performance degrades when encountering data different from its training distribution (Mansour et al., 2009; Ganin et al., 2016; Long et al., 2018; Duchi & Namkoong, 2021). This challenge is a major barrier to deploying safe and robust machine learning systems in the real world. Research to address this has yielded several paradigms. An emerging line of work is based on the concept of *disagreement discrepancy* (Rosenfeld & Garg, 2023). This refers to a measure of how the disagreement between two models—a trainable model and a reference model—changes from a source distribution to a target distribution. Maximizing disagreement discrepancy has proven to be versatile, with applications ranging from bounding model error on unlabeled target data (Rosenfeld & Garg, 2023), to designing statistical tests for harmful shifts (Ginsberg et al., 2023), and training models with improved robustness to distribution shifts (Pagliardini et al., 2023).

While promising, this line of work harbors a critical, unaddressed challenge. The true objective for disagreement discrepancy involves the non-differentiable zero-one loss, making it incompatible with gradient-based optimization. Consequently, all existing methods rely on continuous surrogate losses. This raises a fundamental question that has been overlooked: does optimizing the surrogate objective faithfully optimize the true disagreement discrepancy? This concern is not merely theoretical; Mishra & Liu (2025) reported instabilities during training, suggesting that current surrogates may be ill-suited for the task.

In this work, we provide the first rigorous analysis of surrogates for disagreement discrepancy. We ground our analysis in the framework of *Bayes consistency* (Steinwart, 2007). While stronger guarantees like $\mathcal{H}$-consistency exist (Awasthi et al., 2022a;b), their application to the complex models used in practice remains an open challenge. Bayes consistency is therefore the crucial first step: a surrogate that is not sound in the asymptotic limit is fundamentally unreliable.

Our analysis requires extending the standard framework for surrogate loss consistency, originally developed for classification (Zhang, 2004b), to the unique setting of an objective that is a difference of risks. The core of our theoretical contribution is the development of lower and upper bounds on the objective's optimality gap. These bounds are analogous to classic *excess error bounds* from statistical learning theory (Zhang, 2004a; Bartlett et al., 2006). Using this machinery, we prove that existing surrogates are not Bayes consistent, revealing a foundational flaw in current methods. We then design a novel surrogate objective that combines the standard cross-entropy loss for the source risk with a new disagreement loss that is specifically designed to pair with it. We prove that our surrogate is the first to achieve Bayes consistency for the task of maximizing disagreement discrepancy, establishing a principled foundation for reliably optimizing these objectives in practice.

We empirically validate our surrogate in the context of two important downstream applications. First, we consider bounding a model's error under shift, following the framework of Rosenfeld & Garg (2023). The validity and tightness of this error bound depend directly on accurately estimating the disagreement discrepancy, maximized over a class of critic models. Our experiments, conducted across an array of vision benchmarks, e.g., WILDs (Koh et al., 2021), BREEDs (Santurkar et al., 2021), and DomainNet (Peng et al., 2019), and training methods, demonstrate that our surrogate provides a more accurate estimate of this value, achieving a larger maximized disagreement discrepancy than existing surrogates in almost 80% of scenarios tested. Furthermore, we introduce a challenging new evaluation with adversarially chosen target data, where our surrogate exhibits significantly superior robustness. Second, we consider harmful covariate shift detection (Ginsberg et al., 2023), where we show that our consistent surrogate translates to higher statistical power.

By resolving this foundational inconsistency, our work establishes a more principled and reliable basis for the use of disagreement discrepancy in analyzing and improving model robustness under distribution shift.

## 2 RELATED WORK

Our work on the consistency of surrogates for disagreement discrepancy intersects with two main areas: the study of consistency in machine learning and the development of discrepancy-based methods for addressing distribution shift. Below we review key literature in these areas.

### 2.1 CONSISTENCY IN MACHINE LEARNING

Analysis of surrogate objectives in machine learning centers on ensuring that optimizing a tractable surrogate also optimizes the true, intractable target. This is formalized through a hierarchy of guarantees, each offering a different level of assurance. Our work extends this line of inquiry to the setting of disagreement discrepancy, which combines risks with differing losses and distributions.

The foundational guarantee is *Bayes consistency* (Steinwart, 2007), an asymptotic property requiring that the minimizer of the surrogate objective over all measurable functions is also optimal for the target objective. It has been established for convex margin-based losses in binary classification (Zhang, 2004a; Bartlett et al., 2006) and extended to multi-class settings (Tewari & Bartlett, 2007).

A more refined asymptotic guarantee is $\mathcal{H}$-*consistency* (Awasthi et al., 2022a). Instead of considering all functions, it restricts the analysis to a specific hypothesis set $\mathcal{H}$. It requires that the learned model's target risk converges to the risk of the best model within the set $\mathcal{H}$. Recent work has explored $\mathcal{H}$-consistency for binary and multi-class classification (Awasthi et al., 2022a;b; Mao et al., 2023a), as well as for tasks like pairwise ranking (Mao et al., 2023b), learning with abstention (Mao et al., 2024), and structured prediction (Mao et al., 2023c). However, these studies generally rely on hypothesis sets (e.g., linear models, one-layer networks) that are not representative of the complex, deep neural networks used in modern practice. This limited applicability to practical model classes motivates our focus on Bayes consistency as the crucial first step in our analysis.

To formally establish consistency, we use the powerful tool of the *excess error bound*. This provides a quantitative link between the surrogate and target suboptimality, stating that an $\epsilon$-suboptimal surrogate solution is at most $f(\epsilon)$-suboptimal for the target. As noted by Awasthi et al. (2022a), such a bound is a necessary precursor to any full finite-sample guarantee, as it provides the link between the statistical error (from finite data) and the true target error. Our work establishes this foundational link for

disagreement discrepancy: we prove an excess error bound to establish the Bayes consistency of our proposed surrogate, analogous to the classic result of Zhang (2004a).

## 2.2 DISAGREEMENT DISCREPANCY AND RELATED CONCEPTS

The concept of disagreement discrepancy is closely related to $\mathcal{H}\Delta\mathcal{H}$-divergence (Ben-David et al., 2010), which measures the maximal disagreement between any two models from a class $\mathcal{H}$ across two distributions. While foundational for error bounds under distribution shift, its direct computation is intractable due to the maximization over all model pairs.

Recent work has operationalized this idea by maximizing disagreement discrepancy with respect to only one model (a critic) against a fixed reference model. Rosenfeld & Garg (2023) used this approach to bound test error under distribution shift using unlabeled data, proposing a smooth disagreement loss as a surrogate for maximizing disagreement. Building on this, Mishra & Liu (2025) introduced a discounted disagreement to address potential instabilities where the source and target domains overlap, resulting in less conservative error bounds. Our analysis shows that the surrogate objectives in these works are not Bayes consistent.

The versatility of this critic-based disagreement framework has led to its adoption in other contexts. For detecting harmful distribution shifts, Ginsberg et al. (2023) developed a statistical test using an ensemble of classifiers that maximize out-of-domain disagreement while maintaining in-domain consistency. However, our analysis shows that their surrogate, incorporating a disagreement cross-entropy loss, is also inconsistent. Beyond shift detection, Pagliardini et al. (2023) propose D-BAT, which uses a disagreement discrepancy-based objective as a diversity-inducing regularizer for training ensembles. By encouraging agreement between models on training data but disagreement on out-of-distribution data, they empirically show that the induced diversity can help mitigate shortcut learning and transferability. While empirically successful, the surrogate objective used in their work, like others in the literature, lacks a formal consistency guarantee. Our proposed surrogate provides a direct path to placing such powerful training methods on a more reliable theoretical foundation.

## 3 PROBLEM SETTING AND PRELIMINARIES

To formally analyze surrogates for disagreement discrepancy, we first establish our problem setting. We focus on the covariate shift setting in which the input distribution changes from source to target, while the conditional output distribution remains the same. Formally, we define an input space $\mathcal{X}$ with source and target distributions $S$ and $T$, respectively. The corresponding output space $\mathcal{Y}$ is the set of $K$ classes $[\![K]\!] = \{1, \ldots, K\}$ unless otherwise specified. We consider the case where there is a single ground truth output for each input, represented by a labeling function $y^\star : \mathcal{X} \to \mathcal{Y}$.

For any subset $\mathcal{X}' \subseteq \mathcal{X}$, we use $S|_{\mathcal{X}'}$ to denote the distribution $S$ restricted to $\mathcal{X}'$.[1] We denote the softmax function as $\sigma(\mathbf{s})_c = e^{s_c} / \sum_{c'=1}^{K} e^{s_{c'}}$ for $\mathbf{s} \in \mathbb{R}^K$ and $c \in [\![K]\!]$. The indicator function $\mathbf{1}_A$ returns 1 if predicate $A$ is true and 0 otherwise.

### 3.1 MODELS

Central to the problem formulation is the concept of a critic model, used in recent work (Rosenfeld & Garg, 2023; Ginsberg et al., 2023; Pagliardini et al., 2023) to maximize disagreement discrepancy with respect to fixed reference models. We denote the critic as $f : \mathcal{X} \to \mathcal{Z}$, where typically $\mathcal{Z}$ is the logit space $\mathbb{R}^K$. Reference models, denoted as $h : \mathcal{X} \to \mathcal{Y}$, return raw outputs rather than logits. To accommodate scenarios involving multiple reference models, we use the notation $h = (h_1, h_2, \ldots, h_n)$ when needed. To convert logits to raw outputs, we introduce a utility function $\mathsf{A} : \mathcal{Z} \to \mathcal{Y}$, which, unless otherwise specified, is set to

$$\mathsf{A}(\mathbf{s}) = \min \left( \arg \max_{i \in [\![K]\!]} s_i \right), \tag{1}$$

for $\mathbf{s} \in \mathbb{R}^K$ and we write $\mathsf{A}f(x)$ as shorthand for $\mathsf{A} \circ f(x)$.[2] Here the $\min$ operation is a tie-breaking mechanism, selecting the smallest index where multiple logits share the same maximum value.

---

[1]Formally, for any event $A \subseteq \mathcal{X}$, we define $S|_{\mathcal{X}'}(A) = S(A \cap \mathcal{X}')/S(\mathcal{X}')$ assuming $S(\mathcal{X}') > 0$.

[2]The symbol $\mathsf{A}$ is chosen to represent this function as it alludes to the argmax operation.

## 3.2 Loss Functions and Risk

We consider loss functions of the form $\ell : \mathcal{X} \times \mathcal{Y} \times \mathcal{Z} \to \mathbb{R}$, where $\ell(x, y, z)$ measures the loss for input $x$ with reference model output $y$ and critic model output $z$. Note that while standard losses typically do not depend on $x$, we maintain this general form to accommodate certain analytical loss functions introduced later. Given a loss $\ell$ and reference model $h$, we define the risk of the critic model $f$ on distribution $S$ as $R[\ell, h, f](S) = \mathrm{E}_{X \sim S}[\ell(X, h(X), f(X))]$.

## 3.3 Generalized Disagreement Discrepancy and Surrogates

We now present a generalized formulation of disagreement discrepancy that captures the notions considered in recent work (Rosenfeld & Garg, 2023; Ginsberg et al., 2023; Pagliardini et al., 2023). Disagreement discrepancy quantifies the extent to which a critic model agrees with reference models differently on source and target distributions. Maximizing this measure with respect to the critic has diverse applications, including developing models with robust representations under distribution shift and assessing a model's generalization capability to target distributions.

**Definition 1** (Disagreement Discrepancy). Let $S, T$ be the source and target distributions on input space $\mathcal{X}$. For a pair of reference models $h = (h_1, h_2)$ and a critic model $f$, we define the generalized disagreement discrepancy as

$$\mathrm{d}_\alpha[h, f](S, T) = \alpha R[\ell_{\mathrm{zo}}, h_2, \mathsf{A}f](T) - R[\ell_{\mathrm{zo}}, h_1, \mathsf{A}f](S),$$

where $\ell_{\mathrm{zo}}(x, y, y') = \mathbf{1}_{y \neq y'}$ is the zero-one loss and $\alpha > 0$ allows a trade-off between the two terms. For brevity, we omit the $\alpha$ subscript when $\alpha = 1$.

Previous work has used specific instances of this generalized formulation:

- Rosenfeld & Garg (2023) set $h_1 = h_2$ to the model under evaluation and $\alpha = 1$.
- Ginsberg et al. (2023) set $h_1$ to the ground truth labeling function, $h_2$ to the model under evaluation, and $\alpha \approx 1/N$ where $N$ is the size of the source dataset.
- Pagliardini et al. (2023) set $h_1$ to the ground truth labeling function, $h_2$ to a separate model trained on source data, and treat $\alpha$ as a tunable hyperparameter.

While the disagreement discrepancy is the ideal objective in the above works, its use of the zero-one loss makes it incompatible with gradient-based optimization methods. To address the limitation of non-differentiability, prior work introduced surrogate objectives. We present a generalized formulation of these surrogates that aligns with Definition 1.

**Definition 2** (Surrogate Disagreement Discrepancy). Given loss functions $\ell_{\mathrm{agr}} : \mathcal{X} \times \mathcal{Y} \times \mathcal{Z} \to \mathbb{R}$ and $\ell_{\mathrm{dis}} : \mathcal{X} \times \mathcal{Y} \times \mathcal{Z} \to \mathbb{R}$ differentiable in their third argument, we define a surrogate for generalized disagreement discrepancy as

$$\hat{\mathrm{d}}_\alpha[h, f](S, T) = R[\ell_{\mathrm{agr}}, h_1, f](S) + \alpha R[\ell_{\mathrm{dis}}, h_2, f](T)$$

where $\ell_{\mathrm{agr}}$ encourages agreement and $\ell_{\mathrm{dis}}$ encourages disagreement. This surrogate is designed to be *minimized*, unlike $\mathrm{d}_\alpha[h, f](S, T)$, which is designed to be *maximized*.

Across the literature, the cross-entropy loss

$$\ell_{\mathrm{ce}}(x, y, \mathbf{s}) = -\log \sigma(\mathbf{s})_y \tag{2}$$

has consistently been employed as a surrogate for the agreement loss (Rosenfeld & Garg, 2023; Ginsberg et al., 2023; Pagliardini et al., 2023). In contrast, there is no consensus on a surrogate for the disagreement loss (Chuang et al., 2020; Pagliardini et al., 2023; Ginsberg et al., 2023; Rosenfeld & Garg, 2023). Within this work we focus on the disagreement losses proposed by Rosenfeld & Garg (2023) and Ginsberg et al. (2023):

$$\ell_{\mathrm{dis}}^{\mathrm{RG}}(x, y, \mathbf{s}) = \log\left(1 + \mathrm{e}^{\left(s_y - \frac{1}{K-1}\sum_{c \neq y} s_c\right)}\right), \tag{3}$$

$$\ell_{\mathrm{dis}}^{\mathrm{GLK}}(x, y, \mathbf{s}) = -\frac{1}{K-1}\sum_{c \neq y}\log \sigma(\mathbf{s})_c. \tag{4}$$

While both losses are convex and differentiable in $\mathbf{s}$, we show in Section 4.3 that they do not yield a consistent surrogate for disagreement discrepancy.

## 4 CONSISTENCY OF DISAGREEMENT DISCREPANCY SURROGATES

This section presents our theoretical analysis of surrogate losses for disagreement discrepancy. Consistency is a crucial property for any surrogate objective, as it provides the guarantee that minimizing the surrogate also leads to a solution for the true, non-differentiable objective. This property forms a vital theoretical underpinning for applications that rely on such surrogates. Our analysis proceeds as follows: we first define Bayes consistency for disagreement discrepancy; then we reformulate the disagreement discrepancy to facilitate our analysis; next we prove that existing surrogates are not Bayes consistent; and finally we introduce our novel surrogate and prove its consistency.

### 4.1 BAYES CONSISTENCY FOR DISAGREEMENT DISCREPANCY

Our goal is to determine whether surrogate objectives faithfully optimize the true disagreement discrepancy. To formalize this, we employ the concept of Bayes consistency, a fundamental notion in learning theory that assesses whether optimizing a surrogate loss asymptotically leads to the optimization of the true risk (Zhang, 2004b). We extend this concept to disagreement discrepancy as follows:

**Definition 3** (Bayes consistency for disagreement discrepancy). A surrogate $\hat{\mathrm{d}}_\alpha$ for disagreement discrepancy $\mathrm{d}_\alpha$ is Bayes consistent if, for any sequence of critic models $\{f_n\}$ and distributions $S, T$ on $\mathcal{X}$,

$$\hat{\mathrm{d}}_\alpha[h, f_n](S, T) - \inf_{f' \in \mathcal{H}} \hat{\mathrm{d}}_\alpha[h, f'](S, T) \xrightarrow{p} 0$$

implies

$$\sup_{f' \in \mathcal{H}} \mathrm{d}_\alpha[h, f'](S, T) - \mathrm{d}_\alpha[h, f_n](S, T) \xrightarrow{p} 0.$$

This definition ensures that when a sequence of critic models optimizes the surrogate objective arbitrarily well (in probability), it simultaneously optimizes the true disagreement discrepancy. Bayes consistency provides a theoretical foundation for analyzing surrogate objectives, offering insights into their asymptotic behavior and their relationship to the true disagreement discrepancy. This analysis is particularly valuable given the non-differentiability of the true objective, which precludes direct optimization in practice.

### 4.2 REFORMULATION OF DISAGREEMENT DISCREPANCY

Existing theory for proving consistency of surrogate losses has typically been constructed in terms of objective functions expressible as a *single risk* (Zhang, 2004a; Bartlett et al., 2006; Tewari & Bartlett, 2007). However, the disagreement discrepancy is a sum of *two* risks with respect to *different* distributions, posing a challenge: we cannot simply apply existing theory to each risk separately, as they are intrinsically coupled from an optimization perspective. To overcome this, we present a decomposition that *rewrites* the objective as a sum of decoupled risks using pseudo-losses.

To begin, let $p_S$ and $p_T$ denote the density functions of the source and target distributions, respectively.[3] We define two loss functionals as follows:

$$
\begin{aligned}
L_1[\ell_1, \ell_2](x, \mathbf{y}, z) &= \mathbf{1}_{p_S(x) \geq p_T(x)} \left( \ell_1(x, y_1, z) + \frac{p_T(x)}{p_S(x)} \ell_2(x, y_2, z) \right), \\
L_2[\ell_1, \ell_2](x, \mathbf{y}, z) &= \mathbf{1}_{p_S(x) < p_T(x)} \left( \frac{p_S(x)}{p_T(x)} \ell_1(x, y_1, z) + \ell_2(x, y_2, z) \right),
\end{aligned}
\tag{5}
$$

for $x \in \mathcal{X}$, $\mathbf{y} = (y_1, y_2) \in \mathcal{Y}^2$, $z \in \mathcal{Z}$, and losses $\ell_1, \ell_2 \colon \mathcal{X} \times \mathcal{Y} \times \mathcal{Z} \to \mathbb{R}$, where the dependence on $S$ and $T$ is implicit.

Using these loss functional templates, we rewrite the disagreement discrepancy as

$$\mathrm{d}_\alpha[h, f](S, T) = R[\ell_1, h, \mathsf{A}f](S) + R[\ell_2, h, \mathsf{A}f](T), \tag{6}$$

---

[3]We assume density functions exist with respect to a common dominating measure. The theory generalizes to measure theory by replacing density ratios with Radon-Nikodym derivatives assuming absolute continuity.

and its surrogate as

$$\hat{\mathrm{d}}_\alpha[h, f](S, T) = R[\hat{\ell}_1, h, f](S) + R[\hat{\ell}_2, h, f](T), \tag{7}$$

with pseudo-losses $\ell_1 = L_1[-\ell_{\mathrm{zo}}, \alpha\,\ell_{\mathrm{zo}}]$, $\ell_2 = L_2[-\ell_{\mathrm{zo}}, \alpha\,\ell_{\mathrm{zo}}]$, $\hat{\ell}_1 = L_1[\ell_{\mathrm{agr}}, \alpha\,\ell_{\mathrm{dis}}]$ and $\hat{\ell}_2 = L_2[\ell_{\mathrm{agr}}, \alpha\,\ell_{\mathrm{dis}}]$. (See Appendix B for an analysis of the pointwise optimizers of these pseudo-losses.) This reformulation has a crucial property: for any given input $x \in \mathcal{X}$, precisely one of $\ell_1(x, \mathbf{y}, z)$ and $\ell_2(x, \mathbf{y}, z) = 0$ is non-zero. The same property holds for the surrogate losses $\hat{\ell}_1$ and $\hat{\ell}_2$. This allows pointwise optimization of $f$, effectively decoupling the two risks in our analysis.

In the following subsections, we leverage this reformulation to prove the inconsistency of existing surrogates and introduce a new, consistent surrogate for disagreement discrepancy.

## 4.3 INCONSISTENCY OF PRIOR SURROGATE

Having reformulated the disagreement discrepancy and its surrogate in terms of pseudo-losses, we now analyze the Bayes consistency of existing surrogates. Specifically, we focus on the surrogates proposed by Rosenfeld & Garg (2023) and Ginsberg et al. (2023) and demonstrate they are not Bayes consistent in general. Proofs for this section can be found in Appendices A and C.

Our proof strategy extends the framework of Zhang (2004b) by developing a lower bound on the optimality gap of the true disagreement discrepancy. While Zhang (2004b) provide an upper bound useful for proving Bayes consistency (which we will employ later), our complementary lower bound is crucial for establishing inconsistency. We first develop this lower bound for a single risk in Appendix A and then apply it to disagreement discrepancy in Appendix C.

The inconsistency stems from a fundamental mismatch in the optimal predictions over certain regions of the input space. Specifically, there exist regions where the optimal critic for the surrogate disagrees with the optimal critic for the true disagreement discrepancy. The following theorem formalizes this insight by providing a lower bound on the optimality gap of the true disagreement discrepancy.

**Theorem 4.** *Consider a classification task with $K > 2$ classes, where $h\colon \mathcal{X} \to [\![K]\!]^2$ is a reference model outputting a pair of class labels and $f\colon \mathcal{X} \to \mathbb{R}^K$ is a critic model outputting logits. Let $S, T$ be distributions on $\mathcal{X}$ and $\alpha > 0$. For $\lambda \in (0, 1)$ and $\delta \in (0, \frac{1-\lambda}{2})$, define a restricted input space:*

$$\mathcal{X}' = \left\{ x \in \mathcal{X} : h_1(x) = h_2(x), p_T(x) > 0, \lambda + \delta \leq \frac{p_S(x)}{\alpha p_T(x)} \leq 1 - \delta \right\}, \tag{8}$$

*Let $\hat{\mathrm{d}}_\alpha$ be either Rosenfeld & Garg's surrogate[4] with $\ell_{\mathrm{dis}} = \ell_{\mathrm{dis}}^{\mathrm{RG}}$, $\lambda = K/(2K - 2)$, or Ginsberg et al.'s surrogate with $\ell_{\mathrm{dis}} = \ell_{\mathrm{dis}}^{\mathrm{GLK}}$, $\lambda = 1/(K - 1)$, and $\ell_{\mathrm{agr}} = \ell_{\mathrm{ce}}$ in both cases. Then for both surrogates, there exists a convex function $\zeta : [0, \infty) \to [0, \infty)$ that is continuous at 0 with $\zeta(0) = \delta/(1 - \delta)\mathbf{1}_{S(\mathcal{X}')>0} + \alpha\delta\mathbf{1}_{T(\mathcal{X}')>0}$, such that*

$$\sup_{f' \in \mathcal{H}} \mathrm{d}_\alpha[h, f'](S, T) - \mathrm{d}_\alpha[h, f](S, T) \geq \zeta\left( \hat{\mathrm{d}}_\alpha[h, f](S|_{\mathcal{X}'}, T|_{\mathcal{X}'}) - \inf_{f' \in \mathcal{H}} \hat{\mathrm{d}}_\alpha[h, f'](S|_{\mathcal{X}'}, T|_{\mathcal{X}'}) \right).$$

The key insight of this theorem is that the lower bound function $\zeta$ can be positive at zero, which occurs whenever either the source or target distribution assigns positive measure to the restricted space $\mathcal{X}'$. In these cases, a gap remains between the surrogate and true disagreement discrepancy even when perfectly optimized. This violates the Bayes consistency conditions in Definition 3, yielding the following inconsistency result:

**Corollary 5.** *In the setting of Theorem 4, the surrogates proposed by Rosenfeld & Garg (2023) and Ginsberg et al. (2023) are not Bayes consistent for disagreement discrepancy when $K > 2$.*

This result exposes a fundamental limitation of these surrogates. It shows that there exist distributions for which optimizing them does not optimize the true disagreement discrepancy, even with infinite data and unlimited model capacity. This underscores the need for carefully designed surrogates, as seemingly reasonable choices may yield suboptimal solutions in certain scenarios.

---

[4]We consider a generalization of Rosenfeld & Garg's surrogate with $\alpha > 0$ and distinct reference models.

## 4.4 A New Consistent Surrogate

We now propose a new disagreement loss that, when combined with cross-entropy agreement loss, yields a consistent surrogate for disagreement discrepancy, with proofs contained in Appendix D.

Our analysis of existing surrogates revealed inconsistencies arising from mismatches between the optimal solutions of the surrogate and true objectives. Addressing this issue, we propose the following disagreement loss:

$$\ell_{\text{dis}}^{\text{Ours}}(x, y, \mathbf{s}) = -\log\left(1 - \sigma(\mathbf{s})_y\right). \tag{9}$$

The design of this loss is motivated by its symmetry with the cross-entry agreement loss (2). While minimizing cross-entropy agreement loss $-\log\sigma(\mathbf{s})_y$ encourages agreement with $y$ by setting $\sigma(\mathbf{s})_y = 1$, our disagreement loss $-\log(1 - \sigma(\mathbf{s})_y)$ encourages disagreement with $y$ by setting $\sigma(\mathbf{s})_y = 0$. Importantly, our disagreement loss doesn't specify how the remaining probabilities should be configured, aligning with the true disagreement loss $-\mathbf{1}_{y \neq \mathsf{A}(\mathbf{s})}$. This symmetry and alignment with the true losses contribute to the consistency of our surrogate.

To formally establish consistency, we employ the framework of Zhang (2004b), which provides an upper bound on the optimality gap for a true risk in terms of the optimality gap of a surrogate risk. We extend this result to disagreement discrepancy, leveraging our reformulation of the objective as a sum of two disjoint risks. The resulting upper bound is presented in the following theorem:

**Theorem 6.** *Consider a classification task where $h\colon \mathcal{X} \to [\![K]\!]^2$ is a reference model outputting a pair of class labels and $f\colon \mathcal{X} \to \mathbb{R}^K$ is a critic model outputting logits. For any $\alpha > 0$, let $\hat{\mathrm{d}}_\alpha$ be our surrogate with $\ell_{\text{dis}} = \ell_{\text{dis}}^{\text{Ours}}$ and $\ell_{\text{agr}} = \ell_{\text{ce}}$. Then, for any distributions $S, T$ on $\mathcal{X}$, there exists a concave function $\xi : [0, \infty) \to [0, \infty)$ that is continuous at 0 with $\xi(0) = 0$, such that*

$$\sup_{f \in \mathcal{H}} \mathrm{d}_\alpha[h, f'](S, T) - \mathrm{d}_\alpha[h, f](S, T) \leq \xi\left(\hat{\mathrm{d}}_\alpha[h, f](S, T) - \inf_{f' \in \mathcal{H}} \hat{\mathrm{d}}_\alpha[h, f'](S, T)\right).$$

This result is analogous to the classic *excess error bounds* from statistical learning theory (Zhang, 2004a), adapted here for an objective that is not a traditional risk/error. As discussed in Section 2.1, this type of bound is the crucial component for controlling the *calibration error* in a full finite-sample analysis. The key property of the bounding function $\xi$ in our theorem is that it is continuous at 0 with $\xi(0) = 0$. This property ensures that as the surrogate optimality gap vanishes, so too does the true optimality gap, which directly leads to the following consistency result:

**Corollary 7.** *Our surrogate for disagreement discrepancy with cross-entropy agreement loss and the disagreement loss specified in (9) is Bayes consistent for all $K \geq 2$.*

In guaranteeing that optimizing our surrogate will optimize the true disagreement discrepancy, in the limit of infinite data and unrestricted model capacity, we are able to address the fundamental limitations in prior works. This also provides the theoretical foundation for the use of our surrogate in applications involving disagreement discrepancy.

*Remark* 8. Theorem 6 and Corollary 7 also hold for the surrogates of Rosenfeld & Garg (2023) and Ginsberg et al. (2023) when $K = 2$, as they are equivalent to our surrogate in the binary setting.

## 5 Empirical Evaluation of Surrogates

We evaluate our surrogate on two downstream applications where maximizing disagreement discrepancy is central: bounding model error under covariate shift and detecting harmful distribution shifts. In both cases, the validity of the downstream result depends on accurate optimization of the true disagreement discrepancy.

### 5.1 Application: Error Bounds under Covariate Shift

We first consider the framework of Rosenfeld & Garg (2023) for bounding model error under covariate shift. Their key result is a probabilistic upper bound, composed of three terms: the empirical source error, a sample correction term, and the empirical disagreement discrepancy (see Appendix E).

Crucially, the disagreement discrepancy term is estimated by optimizing a surrogate for disagreement discrepancy, and the *reliability* of the entire bound depends on the quality of this optimization. As we

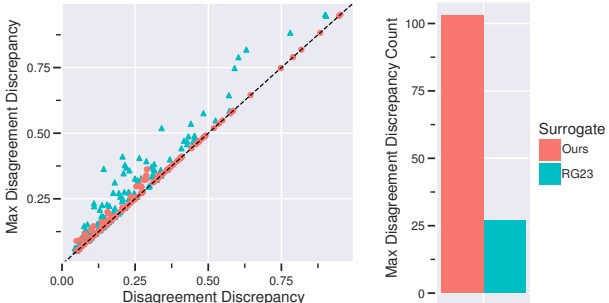 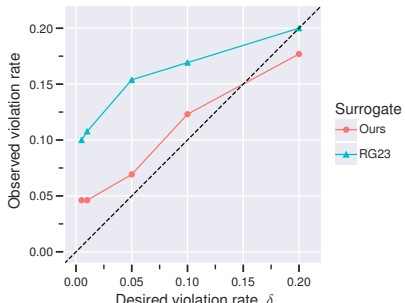

Figure 1: Comparison of disagreement discrepancy estimates for each surrogate. Left: Estimated vs. maximum achieved discrepancy across 130 shifts/models, where proximity to dashed line indicates better performance. Right: Frequency of achieving maximum discrepancy.

Figure 2: Calibration of error bounds: observed vs. desired violation rates ($\delta$). The dashed $y = x$ line represents perfect calibration. Our surrogate demonstrates improved calibration.

will show, underestimating the true disagreement discrepancy—a risk with inconsistent surrogates—yields a deceptively tighter bound that may be *invalid* (i.e., the true error exceeds the bound). This provides a rigorous testbed for our work: better surrogates recover larger discrepancies and yield more trustworthy error bounds.

Our experiments evaluate how different surrogates for disagreement discrepancy affect the error bound's performance, replicating the setup of Rosenfeld & Garg (2023) under natural shifts (Section 5.1.1) and extending it to scenarios with adversarially chosen target data (Section 5.1.2).

### 5.1.1 Replication of Experiments with Existing and New Surrogates

To compare the effectiveness of our surrogate with that of Rosenfeld & Garg (2023), we focus on the disagreement discrepancy term of the error bound mentioned above (see Theorem 21 in Appendix E for details). A larger value of this term indicates a better estimate for a fixed critic hypothesis class $\mathcal{H}$.

We replicated the experiments of Rosenfeld & Garg (2023) using their code, evaluating our proposed surrogate alongside theirs across 11 vision benchmark datasets for distribution shift. Models under evaluation ($h$) were trained on source data using empirical risk minimization (ERM) or one of four unsupervised domain adaptation methods: FixMatch (Sohn et al., 2020), BN-adapt (Li et al., 2017), DANN (Ganin et al., 2016) or CDAN (Long et al., 2018). The critic model $f$ was constructed by appending a tunable linear layer to the frozen weights of $h$, transforming the original logits.

Figure 1 compares the disagreement discrepancies achieved by each surrogate against the maximum across 130 shift and model combinations. Since the true maximum is intractable, we use the largest value achieved by any surrogate in each scenario as a practical reference. Our surrogate attains this maximum in nearly 80% of cases, indicating stronger estimates of the true disagreement discrepancy. A one-sided Wilcoxon signed-rank test also confirms its superiority ($p = 1.8 \times 10^{-11}$).

These results are complemented by Figure 2, which reports the calibration of error bounds for each surrogate. Our surrogate yields better calibration than Rosenfeld & Garg's across most $\delta$ values, producing more reliable bounds, though both methods show elevated violation rates for small $\delta$. Appendix G.1 provides additional comparisons of error bounds versus actual errors, disaggregated by training method and critic architecture.

### 5.1.2 Robustness to Adversarially Chosen Target Data

To further assess the reliability of error bounds, we extend our evaluation to scenarios with adversarially chosen target data—a setting not considered in prior work. This stress test provides crucial insights into how the bounds perform under more challenging conditions.

While adversarially chosen target data was not considered by Rosenfeld & Garg (2023), we still closely following their experimental setup, using 8 of their datasets and focusing on models trained with ERM. We construct adversarial target data by iteratively maximizing the gap between the bound

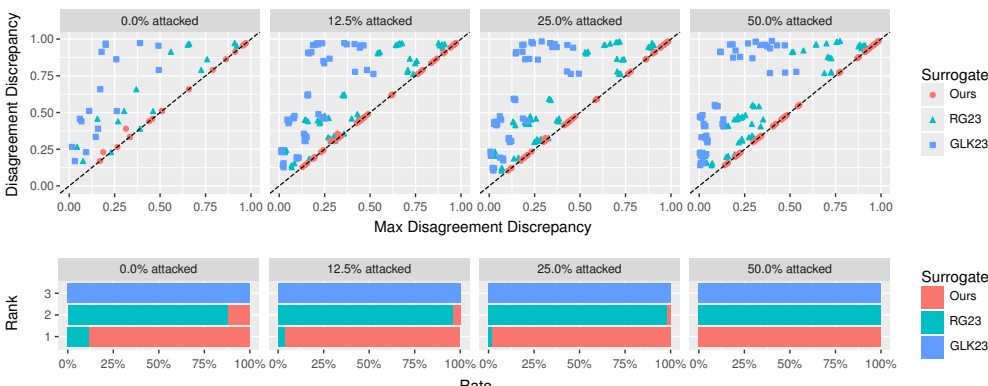

Figure 3: Comparison of disagreement discrepancy estimates for each surrogate under adversarial attacks on target data. Top: Estimated vs. maximum achieved disagreement discrepancy for each surrogate (GLK23, Ours, RG23), faceted by fraction of attacked instances. Points closer to the dashed line indicate better performance. Bottom: Corresponding bar plots displaying the rate at which each surrogate achieves rank 1 (highest), 2, or 3 (lowest) disagreement discrepancy.

and true target error, subject to $\ell_\infty$-norm constraints. Across our experiment set, we test different fractions of attacked data $f$, using values of $0\%$, $12.5\%$, $25\%$ and $50\%$. Further details of our attack procedure and experimental setup can be found in Appendices F and G.2, respectively.

Figure 3 compares our surrogate with those of Rosenfeld & Garg (2023) and Ginsberg et al. (2023) in estimating the maximum disagreement discrepancy under adversarial target data. Results are shown as scatter plots of each surrogate's estimate versus the maximum achieved across surrogates for that scenario, faceted by the attack fraction $f$. Accompanying rank plots show the frequency with which each surrogate achieves the top estimate. Our surrogate exhibits strong robustness, achieving the highest estimate in $87.5\%$ of cases for $f = 0\%$, rising to $100\%$ for $f = 50\%$. A one-sided Wilcoxon signed-rank test against a composite baseline (the best of the RG23 and GLK23 surrogates) confirms this advantage across all $f$, with $p = 3.3 \times 10^{-4}$ at $f = 0\%$ and $p < 6.0 \times 10^{-10}$ for $f > 0\%$. These results indicate that our surrogate provides more reliable discrepancy estimates under adversarial conditions, enabling more robust error bounds. Additional results, including comparisons of true error versus bounds and breakdowns by shift, appear in Appendix G.2.

## 5.2 APPLICATION: DETECTING HARMFUL COVARIATE SHIFT

Maximizing disagreement discrepancy also serves as a powerful mechanism for detecting distribution shift. We explore this application using the *Detectron* framework (Ginsberg et al., 2023), which detects harmful covariate shift for a deployed model $h$ via a hypothesis test. Specifically, a critic model $f$ is trained to maximize disagreement discrepancy with $h$ on unlabeled target data; if the resulting disagreement rate significantly exceeds the rate expected under the source distribution, the shift is flagged as harmful.

To assess the role of surrogate consistency in this setting, we adopt the experimental setup of Ginsberg et al. (2023) on the UCI Heart Disease (HD) dataset (Andras Janosi, 1989). We retain the original 5-class labels (disease severity) rather than the binary target used in prior work, as our surrogate and the baseline coincide when $K = 2$. We train XGBoost (Chen & Guestrin, 2016) critics, comparing our surrogate against the GLK23 baseline (see Appendix G.3 for details). We vary the number of target samples $N \in \{10, 20, 50\}$, and repeat each experiment 500 times to estimate ROC curves. Confidence intervals for the AUC and ROC curves are computed using stratified bootstrapping with 1000 samples.

Figure 4 and Table 1 present the results. Our surrogate consistently outperforms the baseline across all sample sizes. As shown in Table 1, the $95\%$ confidence intervals for the AUC do not overlap between the two methods for any $N$, confirming that the improvement is statistically significant in both low-data and higher-data regimes. These results suggest that for this task, the theoretical consistency of the loss function translates to improved statistical power.

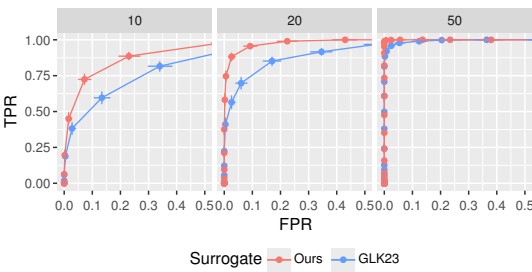

| $N$ | Surrogate | AUC-ROC |
|-----|-----------|---------|
| 10 | GLK23 | $0.821^{+0.025}_{-0.025}$ |
|    | Ours | $0.908^{+0.019}_{-0.017}$ |
| 20 | GLK23 | $0.913^{+0.016}_{-0.019}$ |
|    | Ours | $0.984^{+0.005}_{-0.006}$ |
| 50 | GLK23 | $0.995^{+0.003}_{-0.002}$ |
|    | Ours | $1.000^{+0.000}_{-0.000}$ |

Figure 4: ROC curves for harmful shift detection on UCI-HD. Error bars indicate 95% bootstrapped confidence intervals.

Table 1: AUC-ROC for shift detection on UCI-HD.

## 6 CONCLUSION

Disagreement discrepancy has emerged as a powerful framework for addressing distribution shift, with applications spanning error bounding, shift detection, and robust model training. However, this work reveals a fundamental theoretical flaw: existing surrogate objectives for disagreement discrepancy are not Bayes consistent, meaning that optimizing these surrogates can fail to optimize the true disagreement discrepancy. This inconsistency undermines the theoretical foundations of methods that rely on these surrogates and may explain practical instabilities reported in prior work.

Our theoretical analysis provides both upper and lower bounds on the optimality gap between true and surrogate objectives, establishing a comprehensive framework for understanding surrogate quality in this setting. Guided by this theory, we propose a novel disagreement loss that, when paired with cross-entropy, yields the first provably Bayes consistent surrogate for disagreement discrepancy. Our empirical evaluation demonstrates that this theoretical improvement translates to practical benefits in downstream applications: our surrogate consistently yields more reliable error bounds under covariate shift, particularly under adversarial conditions, and achieves higher statistical power for detecting harmful covariate shifts.

While our focus on Bayes consistency considers optimization over the class of measurable functions, this choice is deliberate and necessary. $\mathcal{H}$-consistency (Zhang & Agarwal, 2020), while theoretically appealing for its consideration of restricted hypothesis classes, remains limited in practice—no successful $\mathcal{H}$-consistency analysis exists for the deep neural networks used in modern applications. Bayes consistency therefore provides an appropriate and achievable foundation for establishing the soundness of surrogate objectives before pursuing more restrictive analyses.

Our work opens several promising directions. A full finite-sample guarantee requires bounding both the *calibration error* (the gap between surrogate and target objectives) and the *estimation error* (arising from finite samples); our bound directly addresses the former. The next challenge is to develop bounds on the estimation error for disagreement discrepancy. Moreover, while our surrogate resolves the consistency issue, assumptions underlying applications like error bounding may not always hold, as demonstrated by our adversarial experiments. This underscores the need for care when deploying these methods in practice.

By establishing the first consistent surrogate for disagreement discrepancy, our work provides a principled theoretical foundation for this important class of methods. This contribution not only resolves existing inconsistencies but also paves the way for more reliable and robust approaches to handling distribution shift in machine learning systems.

### REPRODUCIBILITY STATEMENT

Complete proofs for all theoretical claims, including the inconsistency of existing surrogates (Theorem 4, Corollary 5) and the consistency of our proposed surrogate (Theorem 6, Corollary 7), are provided in Appendices C and D respectively. The general framework for proving inconsistency is detailed in Appendix A. All mathematical assumptions and conditions are explicitly stated within these proofs.

Detailed experimental configurations are provided in Appendix G. For the replication experiments in Section 5.1.1, we utilize the publicly available code and datasets released by Rosenfeld & Garg (2023), with our modifications clearly documented. For the adversarial robustness experiments in Section 5.1.2, we provide complete source code at github.com/ngmarchant/consistent-disagree-discrep, including scripts for dataset downloading and pre-processing, model training, attack implementation (detailed in Appendix F), and generation of all figures and tables. For the harmful shift experiments in Section 5.2, we use the public code released by Ginsberg et al. (2023), with modifications clearly documented in Appendix G.3.

## ACKNOWLEDGMENTS

The authors would like to thank Chris Leckie, Amar Kaur and Paul Montague for their valuable input during the course of this research project. This work was supported by the Australian Defence Science and Technology Group via the Advanced Strategic Capabilities Accelerator program and the University of Melbourne's Research Computing Services and Petascale Campus Initiative.

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

# A    GENERAL FRAMEWORK FOR PROVING INCONSISTENCY

This appendix presents a general framework for proving the inconsistency of surrogate loss functions in the context of a single risk objective. While the main focus of our paper is on objectives that are a sum of two risks over different distributions with different losses (i.e., the disagreement discrepancy), the results developed here for a single risk serve as a crucial building block for our later analysis.

We adapt and extend the *upper bound* of Zhang (2004b, Appendix A) to develop a *lower bound* on the optimality gap of a target objective in terms of the optimality gap of a surrogate objective. This lower bound is crucial for proving inconsistency, as it allows us to show cases where optimizing the surrogate does not necessarily lead to optimizing the true objective.

Our framework considers a true loss function $\ell : \mathcal{X} \times \mathcal{Y} \times \mathcal{Z}_1 \to \mathbb{R}$ and a surrogate loss function $\hat{\ell} : \mathcal{X} \times \mathcal{Y} \times \mathcal{Z}_2 \to \mathbb{R}$. The true objective is to select a critic model $f : \mathcal{X} \to \mathcal{Z}_2$ that minimizes the risk $R[\ell, h, \mathsf{A}f](D)$ and the surrogate objective is to minimize the risk $R[\hat{\ell}, h, f](D)$. Here $\mathsf{A} : \mathcal{Z}_2 \to \mathcal{Z}_1$ is a mapping between model output spaces—e.g., from logits to class labels. We assume the objective is optimized over critic models in a pointwise optimizable hypothesis class $\mathcal{H} = \{f : \mathcal{X} \to \mathcal{Z}_2\}$, which is relevant for Bayes consistency. Central to our analysis is the concept of excess loss, defined for the true loss as $\Delta \ell(x, y, z) = \ell(x, y, z) - \inf_{z' \in \mathcal{Z}_1} \ell(x, y, z')$, and similarly for the surrogate loss with $z, z' \in \mathcal{Z}_2$.

The key idea of our framework is to relate these excess losses through a carefully constructed functional, which forms the basis for our analysis.

**Definition 9.** Define $\Delta G$ as a functional that takes true and surrogate losses $\ell, \hat{\ell}$ as inputs and returns a mapping $\Delta G[\ell, \hat{\ell}] : [0, \infty) \times \mathcal{X} \times \mathcal{Y} \to [0, \infty)$ such that for any $\epsilon \geq 0$, $x \in \mathcal{X}$, $y \in \mathcal{Y}$:

$$\Delta G[\ell, \hat{\ell}](\epsilon, x, y) = \inf_{z \in C[\hat{\ell}](\epsilon, x, y)} \Delta \ell(x, y, \mathsf{A}(z)),$$

where $C[\hat{\ell}](\epsilon, x, y) = \{z \in \mathcal{Z}_2 : \Delta \hat{\ell}(x, y, z) \leq \epsilon\}$. We also overload $\Delta G$ to define another functional that returns a mapping $\Delta G[\ell, \hat{\ell}] : [0, \infty) \to [0, \infty)$ *with only one argument* such that

$$\Delta G[\ell, \hat{\ell}](\epsilon) = \inf_{(x,y) \in \mathcal{X} \times \mathcal{Y}} \Delta G[\ell, \hat{\ell}](\epsilon, x, y).$$

For brevity, we drop the functional arguments $\ell, \hat{\ell}$ when clear from context.

Intuitively, $\Delta G[\ell, \hat{\ell}](\epsilon, x, y)$ gives the smallest possible value of the true excess loss at a point $(x, y)$, considering all outputs $z$ where the surrogate excess loss is at most $\epsilon$. The overloaded version $\Delta G[\ell, \hat{\ell}](\epsilon)$ extends this idea to the entire input space, providing the smallest true excess loss achievable when the surrogate excess loss is bounded by $\epsilon$ everywhere. This functional allows us to analyze how well optimizing the surrogate loss translates to optimizing the true loss, which is crucial for proving inconsistency.

Next, we establish some properties of $\Delta G[\ell, \hat{\ell}]$ that we will use in our analysis.

**Proposition 10.** $\Delta G[\ell, \hat{\ell}]$ *satisfies the following properties:*

1. $\Delta G[\ell, \hat{\ell}](\epsilon) \geq 0$,

2. $\Delta G[\ell, \hat{\ell}](\infty) = 0$,

3. $\Delta G[\ell, \hat{\ell}](\epsilon)$ *is non-increasing over its domain, and*

4. $\Delta G[\ell, \hat{\ell}](\Delta \hat{\ell}(x, y, z)) \leq \Delta \ell(x, y, \mathsf{A}(z))$ *for any $x \in \mathcal{X}$, $y \in \mathcal{Y}$ and $z \in \mathcal{Z}_2$.*

*Proof.* We prove each property below:

1. This follows since $\Delta \ell(x, y, \mathsf{A}(z)) \geq 0$ for all $x \in \mathcal{X}$, $y \in \mathcal{Y}$ and $z \in \mathcal{Z}_2$.

2. When $\epsilon = +\infty$, $x$, $y$ and $z$ are unconstrained, so the minimal excess loss is zero.

3. Increasing $\epsilon$ relaxes the constraint on $x, y$ and $z$, thereby yielding an infimum that is non-increasing.

4. Replacing the constraint set $\{x' \in \mathcal{X}, y' \in \mathcal{Y}, z' \in \mathcal{Z}_2 : \Delta\hat{\ell}(x', y', z') \leq \Delta\hat{\ell}(x, y, z)\}$ by the subset $\{x' \in \mathcal{X}, y' \in \mathcal{Y}, z' \in \mathcal{Z}_2 : \Delta\hat{\ell}(x', y', z') = \Delta\hat{\ell}(x, y, z)\}$ yields the upper bound.

$\square$

The following theorem is central to our framework. It provides a lower bound on the optimality gap of the true risk in terms of the optimality gap of the surrogate risk, mediated by a convex function.

**Theorem 11.** *Let $\zeta(\epsilon)$ be a convex function on $[0, \infty)$ such that $\zeta(\epsilon) \leq \Delta G[\ell, \hat{\ell}](\epsilon)$. Then for any distribution $D$ on $\mathcal{X}$, reference model $h \colon \mathcal{X} \to \mathcal{Y}$, critic model $f \colon \mathcal{X} \to \mathcal{Z}_2$ and mapping $\mathsf{A} \colon \mathcal{Z}_2 \to \mathcal{Z}_1$ we have*

$$\zeta\Big(\operatorname*{E}_{X \sim D} \Delta\hat{\ell}(X, h(X), f(X))\Big) \leq \operatorname*{E}_{X \sim D} \Delta\ell(X, h(X), \mathsf{A}f(X)).$$

*Proof.* We have

$$\begin{aligned}
\zeta\Big(\operatorname*{E}_{X \sim D} \Delta\hat{\ell}(X, h(X), f(X))\Big) &\leq \operatorname*{E}_{X \sim D} \zeta(\Delta\hat{\ell}(X, h(X), f(X))) && \text{by Jensen's inequality} \\
&\leq \operatorname*{E}_{X \sim D} \Delta G[\ell, \hat{\ell}](\Delta\hat{\ell}(X, h(X), f(X))) && \text{by assumption} \\
&\leq \operatorname*{E}_{X \sim D} \Delta\ell(X, h(X), \mathsf{A}f(X)) && \text{by Proposition 10}
\end{aligned}$$

$\square$

This theorem lays the groundwork for proving Bayes inconsistency. The next step in our analysis is to show the existence of a convex function $\zeta(\epsilon)$ that not only satisfies the conditions of the theorem, but also remains positive as the surrogate optimality gap $\epsilon$ approaches zero under some conditions. We construct such a function below and establish its properties.

**Proposition 12.** *Let $\Delta G \colon [0, \infty) \to [0, \infty)$. The function $\zeta_\star(\epsilon) = \sup_{a \leq 0, b \in \mathbb{R}}\{a\epsilon + b | \forall \omega \geq 0, a\omega + b \leq \Delta G(\omega)\}$ satisfies the following properties:*

1. *$\zeta_\star$ is convex.*

2. *$\zeta_\star(\epsilon) \leq \Delta G(\epsilon)$ for all $\epsilon \geq 0$.*

3. *$\zeta_\star$ is non-increasing.*

4. *For any convex function $\zeta$ such that $\zeta(\epsilon) \leq \Delta G(\epsilon)$, $\zeta(\epsilon) \leq \zeta_\star(\epsilon)$.*

5. *Assume there exists $a < 0$ such that $a\epsilon + \Delta G(0) \leq \Delta G(\epsilon)$. Then $\zeta_\star$ is continuous at $0$.*

*Proof.* We prove each property below:

1. The function $\zeta_\star$ is defined as the pointwise supremum of convex functions, hence it is also convex.

2. This follows directly from the definition of $\zeta_\star$.

3. Consider $\epsilon' \geq \epsilon \geq 0$. For any $a \leq 0$ and $b \in \mathbb{R}$ such that $a\omega + b \leq \Delta G(\omega)$ for all $\omega \geq 0$, we have $a\epsilon' + b \leq a\epsilon + b$. Taking the supremum over all such $a$ and $b$, we get $\zeta_\star(\epsilon') \leq \zeta_\star(\epsilon)$.

4. At any $\epsilon \geq 0$, we can find a line that satisfies $a\omega + b \leq \zeta(\omega)$ for all $\omega \geq 0$ and $\zeta(\epsilon) = a\epsilon + b$. Together with the assumption that $\zeta(\epsilon) \leq \Delta G(\epsilon)$, this implies $\zeta(\epsilon) \leq \zeta_\star(\epsilon)$.

5. For any $\epsilon > 0$, let $\delta = -\frac{\epsilon}{2a}$. Then by definition, $\zeta_\star(\delta) \geq a\delta + \Delta G(0)$ (the line $a\delta + \Delta G(0)$ may not be maximal). So $\zeta_\star(0) - \zeta_\star(\delta) = \Delta G(0) - \zeta_\star(\delta) \leq -a\delta = \epsilon/2 < \epsilon$. Thus $\lim_{\epsilon \to 0^+} \zeta_\star(\epsilon) = \zeta_\star(0)$.

$\square$

Building on the properties of $\zeta_\star$ from Proposition 12, we now identify conditions ensuring the convex function $\zeta_\star$ remains positive as the surrogate optimality gap approaches zero.

**Corollary 13.** *Suppose there exists $\epsilon > 0$ such that $\Delta G[\ell, \hat{\ell}](\epsilon) > 0$. Then there exists a convex function $\zeta$ on $[0, \infty)$ that depends only on the loss functions $\ell, \hat{\ell}$ such that $\zeta$ is continuous at zero and $\zeta(0) = \Delta G[\ell, \hat{\ell}](0) > 0$. Moreover*

$$\zeta\Big(\operatorname*{E}_{X \sim D} \Delta\hat{\ell}(X, h(X), f(X))\Big) \leq \operatorname*{E}_{X \sim D} \Delta\ell(X, h(X), \mathsf{A}f(X))$$

*for any distribution $D$ on $\mathcal{X}$, reference model $h\colon \mathcal{X} \to \mathcal{Y}$, critic model $f\colon \mathcal{X} \to \mathcal{Z}_2$ and mapping $\mathsf{A}\colon \mathcal{Z}_2 \to \mathcal{Z}_1$.*

*Proof.* Consider the convex function $\zeta_\star$ defined in Proposition 12. It satisfies the condition $\zeta(\epsilon) \leq \Delta G[\ell, \hat{\ell}](\epsilon)$, hence the inequality follows from Theorem 11.

We now only need to show that $\zeta_\star$ is continuous at zero. Since $\Delta G(\epsilon)$ is positive in some neighborhood of $\epsilon = 0$, is non-increasing, and bounded below by zero, there exists a line $a\epsilon + b$ with $a < 0$ and $b = \Delta G(0)$ such that $a\epsilon + b \leq \Delta G(\epsilon)$ for all $\epsilon \geq 0$. Hence by Property 5 of Proposition 10, $\zeta_\star$ is continuous at zero. $\square$

## B  EXCESS PSEUDO-LOSSES

In this appendix, we evaluate the excess loss for various pseudo-losses introduced in Section 4.2. Recall from Appendix A, that for a given loss $\ell\colon \mathcal{X} \times \mathcal{Y} \times \mathcal{Z}$, we define the excess loss as $\Delta\ell(x, y, z) = \ell(x, y, z) - \inf_{z \in \mathcal{Z}} \ell(x, y, z)$. This analysis is crucial for understanding the behavior of these pseudo-losses and their impact on the consistency of surrogate objectives for disagreement discrepancy in Sections 4.3 and 4.4.

### B.1  TRUE LOSS

We evaluate the excess loss for the two pseudo-losses $\ell_1, \ell_2$ that appear in the true disagreement discrepancy, as formulated in (6). Specifically, for $i \in \{1, 2\}$ we consider the loss function $\ell_i = L_i[\ell_{\mathrm{zo}}, -\alpha\,\ell_{\mathrm{zo}}]$, where $L_i$ is defined in (5), $\ell_{\mathrm{zo}}$ is the zero-one loss and $\alpha > 0$. For an input $x \in \mathcal{X}$, reference model labels $\mathbf{y} \in [\![K]\!]^2$ and critic model label $y \in [\![K]\!]$, we have

$$\inf_{y' \in [\![K]\!]} \ell_1(x, \mathbf{y}, y') = \inf_{y' \in [\![K]\!]} \mathbf{1}_{p_S(x) \geq p_T(x)} \left( \mathbf{1}_{y_1 \neq y'} - \frac{\alpha p_T(x)}{p_S(x)} \mathbf{1}_{y_2 \neq y'} \right)$$

$$= \mathbf{1}_{p_S(x) \geq p_T(x)} \times \begin{cases} 0, & y_1 = y_2 \wedge \frac{\alpha p_T(x)}{p_S(x)} \leq 1, \\ 1 - \frac{\alpha p_T(x)}{p_S(x)}, & y_1 = y_2 \wedge \frac{\alpha p_T(x)}{p_S(x)} > 1, \\ -\frac{\alpha p_T(x)}{p_S(x)}, & y_1 \neq y_2, \end{cases}$$

where the optimizer is $y' = y_1 = y_2$ for the first case, $y' \neq y_1 = y_2$ for the second case and $y' = y_1 \neq y_2$ for the third case. Similarly,

$$\inf_{y' \in [\![K]\!]} \ell_2(x, \mathbf{y}, y') = \inf_{y' \in [\![K]\!]} \mathbf{1}_{p_S(x) < p_T(x)} \left( \frac{p_S(x)}{p_T(x)} \mathbf{1}_{y_1 \neq y'} - \alpha \mathbf{1}_{y_2 \neq y'} \right)$$

$$= \mathbf{1}_{p_S(x) < p_T(x)} \times \begin{cases} 0, & y_1 = y_2 \wedge \frac{p_S(x)}{\alpha p_T(x)} \geq 1, \\ \frac{p_S(x)}{p_T(x)} - \alpha, & y_1 = y_2 \wedge \frac{p_S(x)}{\alpha p_T(x)} < 1, \\ -\alpha, & y_1 \neq y_2, \end{cases}$$

where the optimizer is $y' = y_1 = y_2$ for the first case, $y' \neq y_1 = y_2$ for the second case, and $y' = y_1 \neq y_2$ for the third case.

Thus we have

$$\Delta \ell_1(x, \mathbf{y}, y') = \mathbf{1}_{p_S(x) \geq p_T(x)} \left( \mathbf{1}_{y_1 \neq y'} - \frac{\alpha p_T(x)}{p_S(x)} \mathbf{1}_{y_2 \neq y'} \right.$$
$$\left. - \mathbf{1}_{\frac{\alpha p_T(x)}{p_S(x)} > 1} \mathbf{1}_{y_1 = y_2} \left( 1 - \frac{\alpha p_T(x)}{p_S(x)} \right) + \frac{\alpha p_T(x)}{p_S(x)} \mathbf{1}_{y_1 \neq y_2} \right)$$

$$\Delta \ell_2(x, \mathbf{y}, y') = \mathbf{1}_{p_S(x) < p_T(x)} \left( \frac{p_S(x)}{p_T(x)} \mathbf{1}_{y_1 \neq y'} - \alpha \mathbf{1}_{y_2 \neq y'} \right.$$
$$\left. - \mathbf{1}_{\frac{\alpha p_T(x)}{p_S(x)} > 1} \mathbf{1}_{y_1 = y_2} \left( \frac{p_S(x)}{p_T(x)} - \alpha \right) + \alpha \mathbf{1}_{y_1 \neq y_2} \right)$$

## B.2 OUR SURROGATE

We evaluate the excess loss for the two pseudo-losses that appear in the surrogate for disagreement discrepancy (7) when using our disagreement loss. Specifically, for $i \in \{1, 2\}$ we consider the loss $\hat{\ell}_i = L_i[\ell_{\text{ce}}, \alpha \ell_{\text{dis}}^{\text{Ours}}]$, where $L_i$ is defined in (5), $\ell_{\text{ce}}$ is the cross-entropy loss defined in (2), $\ell_{\text{dis}}^{\text{Ours}}$ is our disagreement loss defined in (9) and $\alpha > 0$.

Since $\hat{\ell}_1$ and $\hat{\ell}_2$ have a similar functional form, we analyze them together by writing for $i \in \{1, 2\}$, $x \in \mathcal{X}, \mathbf{y} \in [\![K]\!]^2$ and $\mathbf{s} \in \mathbb{R}^K$:

$$\hat{\ell}_i(x, \mathbf{y}, \mathbf{s}) = \zeta_i(x) \left( \rho_{i,1}(x) \ell_{\text{ce}}(x, y_1, \mathbf{s}) + \alpha \rho_{i,2}(x) \ell_{\text{dis}}^{\text{Ours}}(x, y_2, \mathbf{s}) \right),$$

where

$$\zeta_i(x) = \begin{cases} \mathbf{1}_{p_S(x) \geq p_T(x)}, & i = 1, \\ \mathbf{1}_{p_S(x) < p_T(x)}, & i = 2, \end{cases} \quad \text{and} \quad \rho_{i,j}(x) = \begin{cases} \frac{p_S(x)}{p_T(x)}, & i = 2 \wedge j = 1, \\ \frac{p_T(x)}{p_S(x)}, & i = 1 \wedge j = 2, \\ 1, & \text{otherwise.} \end{cases} \quad (10)$$

Observe that

$$\inf_{\mathbf{s} \in \mathbb{R}^K} \hat{\ell}_i(x, \mathbf{y}, \mathbf{s}) = \inf_{\mathbf{s} \in \mathbb{R}^K} \zeta_i(x) \left( -\rho_{i,1}(x) \log \left( \frac{e^{s_{y_1}}}{\sum_c e^{s_c}} \right) - \alpha \rho_{i,2}(x) \log \left( 1 - \frac{e^{s_{y_2}}}{\sum_c e^{s_c}} \right) \right)$$
$$= \inf_{\mathbf{q} \in \Lambda_{K-1}} \zeta_i(x) \left( -\rho_{i,1}(x) \log(q_{y_1}) - \alpha \rho_{i,2}(x) \log(1 - q_{y_2}) \right)$$

where the second equality follows by letting $\mathbf{q} := (e^{s_1}, \dots, e^{s_K}) / \sum_c e^{s_c} \in \Lambda_{K-1}$. If $\rho_{i,1}(x) = 0$ or $\rho_{i,2}(x) = 0$ then $\inf_{\mathbf{s} \in \mathbb{R}^K} \hat{\ell}_i(x, \mathbf{y}, \mathbf{s}) = 0$. Otherwise, there are two cases to consider:

- If $y_1 \neq y_2$: the minimum is achieved at $\mathbf{q}$ such that $q_{y_1} = 1$ with all other components equal to zero. This implies $\mathsf{A}(\mathbf{s}) = y_1$ and $\inf_{\mathbf{s} \in \mathbb{R}^K} \hat{\ell}_i(x, \mathbf{y}, \mathbf{s}) = 0$.

- If $y_1 = y_2$: the minimum is achieved at $\mathbf{q}$ such that $q_{y_1} = \frac{\rho_{i,1}(x)}{\rho_{i,1}(x) + \rho_{i,2}(x)} = \left( 1 + \frac{\alpha p_T(x)}{p_S(x)} \right)^{-1}$ with the remaining mass distributed arbitrarily across the other components. This implies $\mathsf{A}(\mathbf{s}) = y_1$ if $\frac{\alpha p_T(x)}{p_S(x)} < 1$ or $\mathsf{A}(\mathbf{s}) \neq y_1$ if $\frac{\alpha p_T(x)}{p_S(x)} > 1$, and

$$\inf_{\mathbf{s} \in \mathbb{R}^K} \hat{\ell}_i(x, \mathbf{y}, \mathbf{s}) = \zeta_i(x) \left( \rho_{i,1}(x) \log \left( 1 + \frac{\alpha p_T(x)}{p_S(x)} \right) + \alpha \rho_{i,2}(x) \log \left( 1 + \frac{p_S(x)}{\alpha p_T(x)} \right) \right).$$

We note that the optimizers in each case match the corresponding optimizers for the true loss.

Thus we have $\Delta\,\hat\ell_2(x,\mathbf{y},\mathbf{s}) = \alpha\log\Big(\frac{\sum_c \mathrm{e}^{s_c}}{\sum_{c\neq y_2}\mathrm{e}^{s_c}}\Big)$ if $p_S(x)=0$ and $\Delta\,\hat\ell_1(x,\mathbf{y},\mathbf{s}) = \log\Big(\frac{\sum_c \mathrm{e}^{s_c}}{\mathrm{e}^{s_{y_1}}}\Big)$ if $p_T(x) = 0$. Otherwise

$$\Delta\,\hat\ell_1(x,\mathbf{y},\mathbf{s}) = -\mathbf{1}_{p_S(x)\geq p_T(x)}\left(\log\left(\frac{\mathrm{e}^{s_{y_1}}}{\sum_c \mathrm{e}^{s_c}}\right) + \frac{\alpha p_T(x)}{p_S(x)}\log\left(1 - \frac{\mathrm{e}^{s_{y_2}}}{\sum_c \mathrm{e}^{s_c}}\right)\right.$$
$$\left. + \mathbf{1}_{y_1=y_2}\left(\log\left(1 + \frac{\alpha p_T(x)}{p_S(x)}\right) + \frac{\alpha p_T(x)}{p_S(x)}\log\left(1 + \frac{p_S(x)}{\alpha p_T(x)}\right)\right)\right),$$

$$\Delta\,\hat\ell_2(x,\mathbf{y},\mathbf{s}) = -\mathbf{1}_{p_S(x)<p_T(x)}\left(\frac{p_S(x)}{p_T(x)}\log\left(\frac{\mathrm{e}^{s_{y_1}}}{\sum_c \mathrm{e}^{s_c}}\right) + \alpha\log\left(1 - \frac{\mathrm{e}^{s_{y_2}}}{\sum_c \mathrm{e}^{s_c}}\right)\right.$$
$$\left. + \mathbf{1}_{y_1=y_2}\left(\frac{p_S(x)}{p_T(x)}\log\left(1 + \frac{\alpha p_T(x)}{p_S(x)}\right) + \alpha\log\left(1 + \frac{p_S(x)}{\alpha p_T(x)}\right)\right)\right).$$

### B.3 ROSENFELD AND GARG'S SURROGATE

We evaluate the excess loss for the two pseudo-losses that appear in the surrogate for disagreement discrepancy (7) when using Rosenfeld & Garg's disagreement loss. Specifically, for $i \in \{1, 2\}$ we consider the loss $\hat\ell_i = L_i[\ell_{\mathrm{ce}}, \alpha\,\ell_{\mathrm{dis}}^{\mathrm{RG}}]$, where $L_i$ is defined in (5), $\ell_{\mathrm{ce}}$ is the cross-entropy loss defined in (2), $\ell_{\mathrm{dis}}^{\mathrm{RG}}$ is Rosenfeld & Garg's disagreement loss defined in (3), $\alpha > 0$ and $K > 2$.

Since $\hat\ell_1$ and $\hat\ell_2$ have a similar functional form, we analyze them together by defining for $i \in \{1, 2\}$, $x \in \mathcal{X}, \mathbf{y} \in [\![K]\!]^2, \mathbf{s} \in \mathbb{R}^K$:

$$\hat\ell_i(x,\mathbf{y},\mathbf{s}) = \zeta_i(x)\left(\rho_{i,1}(x)\,\ell_{\mathrm{ce}}(x,y_1,\mathbf{s}) + \alpha\rho_{i,2}(x)\,\ell_{\mathrm{dis}}^{\mathrm{RG}}(x,y_2,\mathbf{s})\right)$$

where $\zeta_i$ and $\rho_{i,j}$ are defined in (10).

Substituting the expressions for $\ell_{\mathrm{ce}}$ and $\ell_{\mathrm{dis}}^{\mathrm{RG}}$ we have

$$\inf_{\mathbf{s}\in\mathbb{R}^K} \hat\ell_i(x,\mathbf{y},\mathbf{s}) = \inf_{\mathbf{s}\in\mathbb{R}^K} \zeta_i(x)\left(\rho_{i,1}(x)\log\left(\frac{\sum_c \mathrm{e}^{s_c}}{\mathrm{e}^{s_{y_1}}}\right) + \alpha\rho_{i,2}(x)\log\left(1 + \mathrm{e}^{s_{y_2} - \frac{1}{K-1}\sum_{c\neq y_2} s_c}\right)\right).$$
$$(11)$$

For the special cases where $\rho_{i,1}(x) = 0$ or $\rho_{i,2}(x) = 0$, we observe that the minimum loss is zero. For the more general case where $\rho_{i,1}(x) > 0$ and $\rho_{i,2}(x) > 0$, we solve the problem by computing the gradient with respect to $\mathbf{s}$, and solving for the stationary points.

Let $\mathbf{q} = (\mathrm{e}^{s_1}, \ldots, \mathrm{e}^{s_K})/\sum_c \mathrm{e}^{s_c}$ denote the softmax probability vector corresponding to $\mathbf{s}$. There are four distinct cases to consider for the gradient components:

1. For $k = y_1 \neq y_2$, we have

$$\frac{\partial\,\hat\ell_i(x,\mathbf{y},\mathbf{s})}{\partial s_k} = \zeta_i(x)\left(\rho_{i,1}(x)(q_k - 1) - \frac{\alpha\rho_{i,2}(x)}{K-1}\frac{1}{1 + \mathrm{e}^{-s_{y_2} + \frac{1}{K-1}\sum_{c\neq y_2} s_c}}\right)$$

2. For $k = y_1 = y_2$, we have

$$\frac{\partial\,\hat\ell_i(x,\mathbf{y},\mathbf{s})}{\partial s_k} = \zeta_i(x)\left(\rho_{i,1}(x)(q_k - 1) + \alpha\rho_{i,2}(x)\frac{1}{1 + \mathrm{e}^{-s_k + \frac{1}{K-1}\sum_{c\neq k} s_c}}\right)$$

3. For $k = y_2 \neq y_1$, we have

$$\frac{\partial\,\hat\ell_i(x,\mathbf{y},\mathbf{s})}{\partial s_k} = \zeta_i(x)\left(\rho_{i,1}(x)q_k + \alpha\rho_{i,2}(x)\frac{1}{1 + \mathrm{e}^{-s_k + \frac{1}{K-1}\sum_{c\neq k} s_c}}\right)$$

4. For $k \neq y_1, k \neq y_2$, we have

$$\frac{\partial\,\hat\ell_i(x,\mathbf{y},\mathbf{s})}{\partial s_k} = \zeta_i(x)\left(\rho_{i,1}(x)q_k - \frac{\alpha\rho_{i,2}(x)}{K-1}\frac{1}{1 + \mathrm{e}^{-s_{y_2} + \frac{1}{K-1}\sum_{c\neq y_2} s_c}}\right)$$

**Case 1:** $y = y_1 = y_2$: Assuming $\zeta_i(x) \neq 0$, there is a stationary point at $\mathbf{s}^\star$ for $i = 1$ and $i = 2$ such that:

$$\rho_{i,1}(x) \left( \frac{e^{s_y^\star}}{e^{s_y^\star} + \sum_{c \neq y} e^{s_c^\star}} - 1 \right) = -\alpha \rho_{i,2}(x) \frac{1}{1 + e^{-s_y^\star + \frac{1}{K-1} \sum_{c \neq y} s_c^\star}},$$

$$\rho_{i,1}(x) \frac{e^{s_k^\star}}{e^{s_k^\star} + \sum_{c \neq k} e^{s_c^\star}} = \frac{\alpha \rho_{i,2}(x)}{K-1} \frac{1}{1 + e^{-s_y^\star + \frac{1}{K-1} \sum_{c \neq y} s_c^\star}}, \quad \forall k \neq y.$$

The equation for components $k \neq y$ implies $s_k^\star = C$ for some constant $C \in \mathbb{R}$, i.e., all components of $\mathbf{s}^\star$ excluding $s_y$ are equal. Hence the system of equations $K$ can be simplified to one equation in $u^\star := s_y^\star - C$:

$$\rho_{i,1}(x) \frac{K-1}{e^{u^\star} + K - 1} = \alpha \rho_{i,2}(x) \frac{1}{1 + e^{-u^\star}}.$$

The solution for both $i = 1$ and $i = 2$ is $u^\star = \log b_K \left( \frac{p_S(x)}{\alpha p_T(x)} \right)$ where

$$b_K(r) = \frac{1}{2}(r-1)(K-1) + \sqrt{(K-1)r + \frac{1}{4}(K-1)^2(r-1)^2}. \tag{12}$$

This corresponds to a score vector $\mathbf{s}^\star$ such that $s_y^\star = C + \log b_K \left( \frac{p_S(x)}{\alpha p_T(x)} \right)$ and $s_k^\star = C$ for all $k \neq y$.

One can show that the behavior of the solution changes at the critical point $r = \frac{p_S(x)}{\alpha p_T(x)} = \frac{K}{2K-2}$. Specifically,

- for $0 < \frac{p_S(x)}{\alpha p_T(x)} < \frac{K}{2K-2}$ we have $0 < b_K \left( \frac{p_S(x)}{\alpha p_T(x)} \right) < 1$ and the critic predicts $\mathsf{A}(\mathbf{s}^\star) \neq y$,

- for $\frac{p_S(x)}{\alpha p_T(x)} = \frac{K}{2K-2}$ we have $b_K \left( \frac{p_S(x)}{\alpha p_T(x)} \right) = 1$ and the critic predicts $\mathsf{A}(\mathbf{s}^\star) = 1$,

- for $\frac{p_S(x)}{\alpha p_T(x)} > \frac{K}{2K-2}$ we have $b_K \left( \frac{p_S(x)}{\alpha p_T(x)} \right) > 1$ and the critic predicts $\mathsf{A}(\mathbf{s}^\star) = y$.

We note that the optimizer for the surrogate losses does not match the optimizer for the true losses at inputs $x$ such that $\frac{K}{2K-2} < \frac{p_S(x)}{\alpha p_T(x)} < 1$. The minimum surrogate loss for a given $x, \mathbf{y}$ is

$$\inf_{\mathbf{s} \in \mathbb{R}^K} \hat{\ell}_i(x, \mathbf{y}, \mathbf{s}) = \zeta_i(x) \rho_{i,1}(x) \log \left( 1 + \frac{K-1}{b_K \left( \frac{p_S(x)}{\alpha p_T(x)} \right)} \right)$$

$$+ \alpha \zeta_i(x) \rho_{i,2}(x) \log \left( 1 + b_K \left( \frac{p_S(x)}{\alpha p_T(x)} \right) \right). \tag{13}$$

**Case 2:** $y_1 \neq y_2$: Let $\mathbf{q} = (e^{s_1}, \ldots, e^{s_K}) / \sum_c e^c$ denote the softmax probability vector corresponding to $\mathbf{s}$. Assuming $\zeta_i(x) \neq 0$, there is a stationary point at $\mathbf{s}^\star$ for $i = 1$ and $i = 2$ such that:

$$\rho_{i,1}(x)(q_{y_1}^\star - 1) = \frac{\alpha \rho_{i,2}(x)}{K-1} \frac{1}{1 + e^{-s_{y_2} + \frac{1}{K-1} \sum_{c \neq y_2} s_c^\star}},$$

$$\rho_{i,1}(x) q_{y_2}^\star = -\alpha \rho_{i,2}(x) \frac{1}{1 + e^{-s_{y_2}^\star + \frac{1}{K-1} \sum_{c \neq y_2} s_c^\star}},$$

$$\rho_{i,1}(x) q_k^\star = \frac{\alpha \rho_{i,2}(x)}{K-1} \frac{1}{1 + e^{-s_{y_2}^\star + \frac{1}{K-1} \sum_{c \neq y_2} s_c^\star}}, \quad \forall k \neq y_1, k \neq y_2.$$

The last equation for components $k \notin \{y_1, y_2\}$ implies $q_k = C$ for some $C \in (0, 1)$, i.e., all components of $\mathbf{q}$ excluding $q_{y_1}^\star$ and $q_{y_2}^\star$ are equal. Using this result, the system of equations simplifies to $(K-1)(1 - q_{y_1}^\star) = q_{y_2}^\star = -(K-1)C$. The only valid solution is obtained in the limit $q_{y_2}^\star = C \to 0$ and $q_{y_1}^\star \to 1$. This implies $\mathsf{A}(\mathbf{s}^\star) = y_1$, which matches the optimizer for the true loss. By appropriately taking the limit, we find the infimum for a given $x, \mathbf{y}$ is therefore

$$\inf_{\mathbf{s} \in \mathbb{R}^K} \hat{\ell}_i(x, \mathbf{y}, \mathbf{s}) = 0$$

Thus we have $\Delta \hat{\ell}_1(x, \mathbf{y}, \mathbf{s}) = \log\left(\frac{\sum_{c'} e^{s_{c'}}}{e^{s_{y_1}}}\right)$ if $p_T(x) = 0$ and $\Delta \hat{\ell}_2(x, \mathbf{y}, \mathbf{s}) = \alpha \log\left(1 + e^{s_{y_2} - \frac{1}{K-1}\sum_{c' \neq y_2} s_{c'}}\right)$ if $p_S(x) = 0$. Otherwise

$$\Delta \hat{\ell}_1(x, \mathbf{y}, \mathbf{s}) = \mathbf{1}_{p_S(x) \geq p_T(x)} \left( \log\left(\frac{\sum_{c'} e^{s_{c'}}}{e^{s_{y_1}}}\right) + \frac{\alpha p_T(x)}{p_S(x)} \log\left(1 + e^{s_{y_2} - \frac{1}{K-1}\sum_{c' \neq y_2} s_{c'}}\right) \right.$$

$$- \mathbf{1}_{y_1 = y_2} \log\left(b_K\left(\frac{p_S(x)}{\alpha p_T(x)}\right) + K - 1\right) - \mathbf{1}_{y_1 = y_2} \log\left(b_K\left(\frac{p_S(x)}{\alpha p_T(x)}\right)\right)$$

$$\left. - \mathbf{1}_{y_1 = y_2} \frac{\alpha p_T(x)}{p_S(x)} \log\left(1 + b_K\left(\frac{p_S(x)}{\alpha p_T(x)}\right)\right) \right),$$

$$\Delta \hat{\ell}_2(x, \mathbf{y}, \mathbf{s}) = \mathbf{1}_{p_S(x) < p_T(x)} \alpha \left( \frac{p_S(x)}{\alpha p_T(x)} \log\left(\frac{\sum_{c'} e^{s_{c'}}}{e^{s_{y_1}}}\right) + \log\left(1 + e^{s_{y_2} - \frac{1}{K-1}\sum_{c' \neq y_2} s_{c'}}\right) \right.$$

$$- \mathbf{1}_{y_1 = y_2} \frac{p_S(x)}{\alpha p_T(x)} \log\left(b_K\left(\frac{p_S(x)}{\alpha p_T(x)}\right) + K - 1\right)$$

$$- \mathbf{1}_{y_1 = y_2} \frac{p_S(x)}{\alpha p_T(x)} \log\left(b_K\left(\frac{p_S(x)}{\alpha p_T(x)}\right)\right)$$

$$\left. - \mathbf{1}_{y_1 = y_2} \log\left(1 + b_K\left(\frac{p_S(x)}{\alpha p_T(x)}\right)\right) \right).$$

### B.4 GINSBERG ET AL.'S SURROGATE

We evaluate the excess loss for the two pseudo-losses that appear in the surrogate for disagreement discrepancy (7) when using Ginsberg et al.'s disagreement loss. Specifically, for $i \in \{1, 2\}$ we consider the loss $\hat{\ell}_i = L_i[\ell_{ce}, \alpha \ell_{dis}^{GLK}]$, where $L_i$ is defined in (5), $\ell_{ce}$ is the cross-entropy loss defined in (2), $\ell_{dis}^{GLK}$ is Ginsberg et al.'s disagreement loss defined in (4), $\alpha > 0$, and $K > 2$.

Since $\hat{\ell}_1$ and $\hat{\ell}_2$ have a similar functional form, we analyze them together by defining for $i \in \{1, 2\}$, $x \in \mathcal{X}, \mathbf{y} \in [\![K]\!]^2, \mathbf{s} \in \mathbb{R}^K$:

$$\hat{\ell}_i(x, \mathbf{y}, \mathbf{s}) = \zeta_i(x) \left( \rho_{i,1}(x) \ell_{ce}(x, y_1, \mathbf{s}) + \alpha \rho_{i,2}(x) \ell_{dis}^{GLK}(x, y_2, \mathbf{s}) \right)$$

$$= \zeta_i(x) \left( \rho_{i,1}(x) \log\left(\frac{\sum_c e^{s_c}}{e^{s_{y_1}}}\right) + \frac{\alpha \rho_{i,2}(x)}{K-1} \sum_{c \neq y_2} \log\left(\frac{\sum_{c'} e^{s_{c'}}}{e^{s_c}}\right) \right)$$

$$= \zeta_i(x) \left( -\rho_{i,1}(x) s_{y_1} - \frac{\alpha \rho_{i,2}(x)}{K-1} \sum_{c \neq y_2} s_c + (\rho_{i,1}(x) + \alpha \rho_{i,2}(x)) \log\left(\sum_c e^{s_c}\right) \right).$$

where $\zeta_i$ and $\rho_{i,j}$ are defined in (10) and we have used the definitions of $\ell_{ce}$ and $\ell_{dis}^{GLK}$ in (2) and (4), respectively.

We now consider the problem of minimizing $\hat{\ell}_i(x, \mathbf{y}, \mathbf{s})$ with respect to $\mathbf{s} \in \mathbb{R}^K$. For the special cases where $\rho_{i,1}(x) = 0$ or $\rho_{i,2}(x) = 0$, we observe that $\inf_{\mathbf{s} \in \mathbb{R}^K} \hat{\ell}_i(x, \mathbf{y}, \mathbf{s}) = 0$. For the more general case where $\rho_{i,1}(x) > 0$ and $\rho_{i,2}(x) > 0$, we solve the problem by computing the gradient with respect to $\mathbf{s}$, and solving for the stationary points.

Let $\mathbf{q} := (e^{s_1}, \ldots, e^{s_K})/\sum_c e^{s_c}$ denote the softmax probability vector corresponding to $\mathbf{s}$. There are four distinct cases to consider for the gradient components:

1. For $k = y_1 \neq y_2$, we have

$$\frac{\partial \hat{\ell}_i(x, \mathbf{y}, \mathbf{s})}{\partial s_k} = \zeta_i(x) \left( -\rho_{i,1}(x) - \frac{\alpha \rho_{i,2}(x)}{K-1} + (\rho_{i,1}(x) + \alpha \rho_{i,2}(x)) q_k \right)$$

2. For $k = y_1 = y_2$, we have

$$\frac{\partial \hat{\ell}_i(x, \mathbf{y}, \mathbf{s})}{\partial s_k} = \zeta_i(x) \left( -\rho_{i,1}(x) + (\rho_{i,1}(x) + \alpha \rho_{i,2}(x)) q_k \right)$$

3. For $k = y_2 \neq y_1$, we have

$$\frac{\partial \hat{\ell}_i(x, \mathbf{y}, \mathbf{s})}{\partial s_k} = \zeta_i(x) \left( \rho_{i,1}(x) + \alpha \rho_{i,2}(x) \right) q_k$$

4. For $k \neq y_1, k \neq y_2$, we have

$$\frac{\partial \hat{\ell}_i(x, \mathbf{y}, \mathbf{s})}{\partial s_k} = \zeta_i(x) \left( -\frac{\alpha \rho_{i,2}(x)}{K - 1} + \left( \rho_{i,1}(x) + \alpha \rho_{i,2}(x) \right) q_k \right)$$

To solve for the stationary points, we consider the cases $y_1 = y_2$ and $y_1 \neq y_2$ separately. Below we define $r(x) := p_S(x)/(\alpha p_T(x))$.

**Case 1:** $y = y_1 = y_2$: Assuming $\zeta_i(x) \neq 0$, there is a stationary point at $\mathbf{q}^\star$ for $i = 1$ and $i = 2$ such that:

$$q_y^\star = \frac{\rho_{i,1}(x)}{\rho_{i,1}(x) + \alpha \rho_{i,2}(x)} = \frac{r(x)}{r(x) + 1},$$

$$q_k^\star = \frac{\frac{\alpha \rho_{i,2}(x)}{K-1}}{\rho_{i,1}(x) + \alpha \rho_{i,2}(x)} = \frac{1}{K - 1} \frac{1}{r(x) + 1}, \quad \forall k \neq y.$$

It is straightforward to verify that this point is a minimizer. This implies $\mathsf{A}(\mathbf{s}^\star) = y$ if $r(x) > \frac{1}{K-1}$ and $\mathsf{A}(\mathbf{s}^\star) \neq y$ if $r(x) < \frac{1}{K-1}$. We note that the optimizer for the surrogate losses does not match the optimizer for the true losses at $x$ such that $\frac{1}{K-1} < \frac{p_S(x)}{\alpha p_T(x)} < 1$. The minimum surrogate loss for a given $x, \mathbf{y}$ is therefore

$$\inf_{\mathbf{s} \in \mathbb{R}^K} \hat{\ell}_i(x, \mathbf{y}, \mathbf{s}) = \rho_{i,1}(x) \log\left( 1 + \frac{\alpha p_T(x)}{p_S(x)} \right)$$

$$+ \alpha \rho_{i,2}(x) \log\left( (K - 1) \left( \frac{p_S(x)}{\alpha p_T(x)} + 1 \right) \right). \tag{14}$$

**Case 2:** $y_1 \neq y_2$: Assuming $\zeta_i(x) \neq 0$, there is a stationary point at $\mathbf{q}^\star$ for $i = 1$ and $i = 2$ such that:

$$q_{y_1}^\star = \frac{\rho_{i,1}(x) + \frac{\alpha \rho_{i,2}(x)}{K-1}}{\rho_{i,1}(x) + \alpha \rho_{i,2}(x)} = \frac{1}{K - 1} \frac{r(x)(K - 1) + 1}{r(x) + 1},$$

$$q_{y_2}^\star = 0,$$

$$q_k^\star = \frac{\frac{\alpha \rho_{i,2}(x)}{K-1}}{\rho_{i,1}(x) + \alpha \rho_{i,2}(x)} = \frac{1}{K - 1} \frac{1}{r(x) + 1}, \quad \forall k \neq y_1, k \neq y_2.$$

It is straightforward to verify that this point is a minimizer. This implies $\mathsf{A}(\mathbf{s}^\star) = y_1$, which matches the optimizer for the true loss. The minimum surrogate loss for a given $x, \mathbf{y}$ is therefore

$$\inf_{\mathbf{s} \in \mathbb{R}^K} \hat{\ell}_i(x, \mathbf{y}, \mathbf{s}) = \left( \rho_{i,1}(x) + \alpha \rho_{i,2}(x) \right) \log\left( (K - 1) \left( \frac{p_S(x)}{\alpha p_T(x)} + 1 \right) \right)$$

$$- \left( \rho_{i,1}(x) + \frac{\alpha \rho_{i,2}(x)}{K - 1} \right) \log\left( (K - 1) \frac{p_S(x)}{\alpha p_T(x)} + 1 \right).$$

Thus we have $\Delta \hat{\ell}_2(x, \mathbf{y}, \mathbf{s}) = \frac{\alpha}{K-1} \sum_{c \neq y_2} \log\left(\frac{\sum_{c'} e^{s_{c'}}}{e^{s_c}}\right)$ if $p_S(x) = 0$ and $\Delta \hat{\ell}_1(x, \mathbf{y}, \mathbf{s}) = \log\left(\frac{\sum_c e^{s_c}}{e^{s_{y_1}}}\right)$ if $p_T(x) = 0$. Otherwise

$$\Delta \hat{\ell}_1(x, \mathbf{y}, \mathbf{s}) = -\mathbf{1}_{p_S(x) \geq p_T(x)} \left( \left(1 + \frac{\mathbf{1}_{y_1 \neq y_2} \alpha p_T(x)}{(K-1)p_S(x)}\right) \log\left(\frac{e^{s_{y_1}}}{\sum_c e^{s_c}} \frac{(K-1)\left(\frac{p_S(x)}{\alpha p_T(x)} + 1\right)}{(K-1)\frac{p_S(x)}{\alpha p_T(x)} + 1}\right) \right.$$

$$\left. + \frac{\alpha p_T(x)}{(K-1)p_S(x)} \sum_{c \notin \{y_1, y_2\}} \log\left(\frac{e^{s_c}}{\sum_{c'} e^{s_{c'}}}(K-1)\left(\frac{p_S(x)}{\alpha p_T(x)} + 1\right)\right) \right),$$

$$\Delta \hat{\ell}_2(x, \mathbf{y}, \mathbf{s}) = -\mathbf{1}_{p_S(x) < p_T(x)} \left( \left(\frac{p_S(x)}{p_T(x)} + \frac{\mathbf{1}_{y_1 \neq y_2} \alpha}{K-1}\right) \log\left(\frac{e^{s_{y_1}}}{\sum_c e^{s_c}} \frac{(K-1)\left(\frac{p_S(x)}{\alpha p_T(x)} + 1\right)}{(K-1)\frac{p_S(x)}{\alpha p_T(x)} + 1}\right) \right.$$

$$\left. + \frac{\alpha}{K-1} \sum_{c \notin \{y_1, y_2\}} \log\left(\frac{e^{s_c}}{\sum_{c'} e^{s_{c'}}}(K-1)\left(\frac{p_S(x)}{\alpha p_T(x)} + 1\right)\right) \right).$$

## C  PROOFS FOR SECTION 4.3

This appendix contains proofs for the results presented in Section 4.3. The key result of this section is Theorem 4, which we prove using Corollary 13 developed in Appendix A.

As a first step, we must prove that the condition for Corollary 13 holds: namely that the relevant $\Delta G$ functional is positive within a neighborhood of zero. This is done for the pseudo-losses associated with the surrogate of Rosenfeld & Garg (2023) below.

**Lemma 14.** *Consider the framework of Appendix A for a classification task with $K > 2$ classes, where $\mathcal{Y} = [\![K]\!]^2$ is the reference output space, $\mathcal{Z}_1 = [\![K]\!]$ is the model output space, and $\mathcal{Z}_2 = \mathbb{R}^K$ is the raw (logit) model output space. Assume the mapping $\mathsf{A} \colon \mathbb{R}^K \to [\![K]\!]$ from logits to predictions is as defined in (1). For fixed $\delta \in \left(0, \frac{K-2}{2K-2}\right)$, set the input space to the restricted input space $\mathcal{X}'$ for Rosenfeld & Garg's surrogate as defined in (8). Then for true loss $\ell_i = L_i[\ell_{\mathrm{zo}}, -\alpha \ell_{\mathrm{zo}}]$ and surrogate loss $\hat{\ell}_i = L_i[\ell_{\mathrm{ce}}, \alpha \ell_{\mathrm{dis}}^{\mathrm{GLK}}]$ we have*

$$\Delta G[\ell_i, \hat{\ell}_i](\epsilon) = \inf_{(x,y) \in \mathcal{X}' \times [\![K]\!]} \Delta G[\ell_i, \hat{\ell}_i](\epsilon, x, y) = \begin{cases} \frac{\delta}{1-\delta}, & i = 1, \\ \alpha\delta, & i = 2, \end{cases}$$

*for all $\epsilon < \epsilon_i^\star(\delta)$ where*

$$\epsilon_i^\star(\delta) = \begin{cases} \log\left(\frac{Kb_K\left(\frac{K}{2K-2}+\delta\right)}{b_K\left(\frac{K}{2K-2}+\delta\right)+K-1}\right) + \frac{2K-2}{K+2\delta(K-1)} \log\left(\frac{2}{1+b_K\left(\frac{K}{2K-2}+\delta\right)}\right), & i = 1, \\ \alpha \frac{K+2\delta(K-1)}{2K-2} \log\left(\frac{Kb_K\left(\frac{K}{2K-2}+\delta\right)}{b_K\left(\frac{K}{2K-2}+\delta\right)+K-1}\right) + \alpha \log\left(\frac{2}{1+b_K\left(\frac{K}{2K-2}+\delta\right)}\right), & i = 2, \end{cases}$$

*and $b_K \colon [0, \infty) \to [0, \infty)$ is defined in (12).*

*Proof.* Let $r(x) = \frac{p_S(x)}{\alpha p_T(x)}$ and let $\zeta_i(x)$ and $\rho_{i,j}(x)$ be as defined in (10). Fix $x \in \mathcal{X}'$ such that $\zeta_i(x) \neq 0$ and $\mathbf{y} = (y, y)$ for $y \in [\![K]\!]$ (i.e., the reference outputs are identical). We begin by proving the following:

Claim: We have

$$\Delta G[\ell_i, \hat{\ell}_i](\epsilon, x, \mathbf{y}) = \begin{cases} r(x)^{-1} - 1, & i = 1, \\ \alpha(1 - r(x)), & i = 2. \end{cases} \tag{15}$$

for all $\epsilon < \epsilon_i^\star(x)$ where

$$\epsilon^\star(x) = \rho_{i,1}(x) \log\left(\frac{Kb_K(r(x))}{b_K(r(x)) + K - 1}\right) + \alpha\rho_{i,2}(x) \log\left(\frac{2}{1 + b_K(r(x))}\right).$$

To prove the claim, we first recall from Definition 9 that:

$$\Delta G[\ell_i, \hat{\ell}_i](\epsilon, x, \mathbf{y}) = \inf_{\mathbf{s} \in C[\hat{\ell}_i](\epsilon, x, \mathbf{y})} \Delta \ell_i(x, \mathbf{y}, \mathsf{A}(\mathbf{s})),$$

with

$$C[\hat{\ell}_i](\epsilon, x, \mathbf{y}) = \{\mathbf{s} \in \mathbb{R}^K : \Delta \hat{\ell}_i(x, \mathbf{y}, \mathbf{s}) \leq \epsilon\}.$$

When $\epsilon = 0$, we know from the analysis in Appendix B.3 that $C[\hat{\ell}_i](0, x, \mathbf{y}) = \{\mathbf{s} \in \mathbb{R}^K : [\forall k \neq y, s_k = C] \wedge [s_y = C + \log b_K(r(x))] \wedge [C \in \mathbb{R}]\}$. For any $\mathbf{s}$ in this set, we have $\mathsf{A}(\mathbf{s}) = y$. It is then straightforward to show, using the expressions for $\Delta \ell_i$ derived in Appendix B.1, that $\Delta G[\ell_i, \hat{\ell}_i](0, x, \mathbf{y})$ is equal to the RHS of (15).

The claim follows if we can prove that $\mathsf{A}(\mathbf{s}) = y$ for all $\mathbf{s} \in C[\hat{\ell}_i](\epsilon, x, \mathbf{y})$ such that $\epsilon < \epsilon^\star(x)$, since the value of $\Delta \ell_i(\epsilon, x, \mathsf{A}(\mathbf{s}))$ and hence $\Delta G[\ell_i, \hat{\ell}_i](\epsilon, x, \mathbf{y})$ are the same as at $\epsilon = 0$. To demonstrate this, we find the minimum surrogate loss $\Delta \hat{\ell}_i(x, y, \mathbf{s})$ with respect to $\mathbf{s} \in \mathbb{R}^K$ such that $\mathsf{A}(\mathbf{s}) \neq y$, and show that it is equal to $\epsilon^\star(x)$. The loss minimization problem is

$$\inf_{\mathbf{s} \in \mathbb{R}^K : \mathsf{A}(\mathbf{s}) \neq y} \Delta \hat{\ell}_i(x, \mathbf{y}, \mathbf{s}) = \inf_{\mathbf{s} \in \mathbb{R}^K : \mathsf{A}(\mathbf{s}) \neq y} \hat{\ell}_i(x, \mathbf{y}, \mathbf{s}) - \inf_{\mathbf{s} \in \mathbb{R}^K} \hat{\ell}_i(x, \mathbf{y}, \mathbf{s}).$$

The unconstrained problem (second term) is solved in Appendix B.3, where minimizers $\mathbf{s}^\star$ are found to satisfy satisfy $s_y^\star > \max_{c \neq y} s_c^\star$. Such minimizers are outside the feasible region for the constrained problem (first term), where we need $s_y^\star < \max_{c \neq y} s_c^\star$. Since the objective is convex in $\mathbf{s}$, solutions to the constrained problem must lie on the boundary where $s_y = \max_{c \neq y} s_c$. This, together with the symmetry of the objective with respect to components $s_k$ for $k \neq y$, means the minimizers $\mathbf{s}^\star$ for the unconstrained problem are vectors where all components are equal.

Substituting the optimizer into the first term above, and using the previously evaluated result (13) for the second term, we have

$$\inf_{\mathbf{s} \in \mathbb{R}^K : \mathsf{A}(\mathbf{s}) \neq y} \Delta \hat{\ell}_2(x, y, \mathbf{s}) = \rho_{i,1}(x) \log K + \alpha \rho_{i,2}(x) \log 2 - \rho_{i,1}(x) \log \left( 1 + \frac{K-1}{b_K \left( \frac{p_S(x)}{\alpha p_T(x)} \right)} \right)$$

$$- \alpha \rho_{i,2}(x) \log \left( 1 + b_K \left( \frac{p_S(x)}{\alpha p_T(x)} \right) \right)$$

$$= \epsilon^\star(x),$$

which completes the proof of the claim.

Next, we find a threshold $\epsilon^\star(\delta)$ that is valid for all $x \in \mathcal{X}'$ by minimizing $\epsilon^\star(x)$ over $x \in \mathcal{X}'$. Since $\epsilon^\star(x)$ only depends on $x$ via $r(x)$, and is a monotonically increasing function of $r(x)$ for $\frac{K}{2K-2} + \delta \leq r(x) \leq 1 - \delta$, we have that the minimum is achieved at $r(x) = \frac{K}{2K-2} + \delta$.

Now by Definition 9 we have for $\epsilon < \epsilon^\star(\delta)$ that

$$\Delta G[\ell_2, \hat{\ell}_2](\epsilon) = \inf_{(x,y) \in \mathcal{X}' \times [\![K]\!]} \Delta G[\ell_2, \hat{\ell}_2](\epsilon, x, y)$$

$$= \inf_{x \in \mathcal{X}'} \alpha(1 - r(x))$$

$$= \alpha \delta.$$

The second inequality follows from the claim proved above and the third inequality follows by setting $r(x) = 1 - \delta$. $\qquad\square$

We obtain a similar result for the surrogate of Ginsberg et al. (2023) below. The proof follows the same structure as the proof of Lemma 14.

**Lemma 15.** *Consider the framework of Appendix A for a classification task with $K > 2$ classes, where $\mathcal{Y} = [\![K]\!]^2$ is the reference output space, $\mathcal{Z}_1 = [\![K]\!]$ is the model output space, and $\mathcal{Z}_2 = \mathbb{R}^K$ is the raw (logit) model output space. Assume the mapping $\mathsf{A} : \mathbb{R}^K \to [\![K]\!]$ from logits to predictions is as defined in (1). For fixed $\delta \in \left( 0, \frac{K-2}{2K-2} \right)$, set the input space to the restricted input space $\mathcal{X}'$ for*

*Ginsberg et al.'s surrogate as defined in (8). Then for true loss $\ell_i = L_i[\ell_{\mathrm{zo}}, -\alpha\,\ell_{\mathrm{zo}}]$ and surrogate loss $\hat{\ell}_i = L_i[\ell_{\mathrm{ce}}, \alpha\,\ell_{\mathrm{dis}}^{\mathrm{GLK}}]$ we have*

$$\Delta G[\ell_i, \hat{\ell}_i](\epsilon) = \inf_{(x,y)\in\mathcal{X}'\times[\![K]\!]} \Delta G[\ell_i, \hat{\ell}_i](\epsilon, x, y) = \begin{cases} \frac{\delta}{1-\delta}, & i = 1, \\ \alpha\delta, & i = 2, \end{cases}$$

*for all $\epsilon < \epsilon_i^\star(\delta)$ where*

$$\epsilon_i^\star(\delta) = \begin{cases} \frac{K-1}{\delta(K-1)+1}\log\left(\frac{K}{\delta(K-1)+K}\right) + \log\left(\frac{\delta K(K-1)+K}{\delta(K-1)+K}\right), & i = 1, \\ \alpha\log\left(\frac{K}{\delta(K-1)+K}\right) + \alpha\frac{\delta(K-1)+1}{K-1}\log\left(\frac{\delta K(K-1)+K}{\delta(K-1)+K}\right), & i = 2. \end{cases}$$

Next we present a result that allows us to compose the convex envelope functions that appear in Appendix A. This is needed, as we will apply the bound of Appendix A to each risk term in the reformulated disagreement discrepancy.

**Lemma 16.** *Let $\zeta_1, \zeta_2\colon [0,\infty) \to [0,\infty)$ be convex functions that are continuous at zero and non-increasing. Then the function $\zeta\colon [0,\infty) \to [0,\infty)$ such that*

$$\zeta(\epsilon) = \inf_{\epsilon_1+\epsilon_2=\epsilon,\epsilon_1\geq 0,\epsilon_2\geq 0} \zeta_1(\epsilon_1) + \zeta_2(\epsilon_2)$$

*has the following properties:*

1. *it satisfies $\zeta(\epsilon_1 + \epsilon_2) \leq \zeta_1(\epsilon_1) + \zeta_2(\epsilon_2)$ for any $\epsilon_1, \epsilon_2 \in [0,\infty)$,*

2. *it is convex,*

3. *it is non-increasing,*

4. *it satisfies $\zeta(0) = \zeta_1(0) + \zeta_2(0)$, and*

5. *it is continuous at zero.*

*Proof.* We prove each property below:

1. This holds trivially by definition.

2. Let $\epsilon, \epsilon' \geq 0$ and $(\epsilon_1, \epsilon_2)$ and $(\epsilon_1', \epsilon_2')$ be pairs that are arbitrarily close to achieving the infimum for $\zeta(\epsilon)$ and $\zeta(\epsilon')$, respectively. For $\lambda \in [0,1]$, the convex combination of these pairs $\lambda(\epsilon_1, \epsilon_2) + (1-\lambda)(\epsilon_1', \epsilon_2')$ has components that sum to $\lambda\epsilon + (1-\lambda)\epsilon'$. By the definition of $\zeta$, we have $\zeta(\lambda\epsilon + (1-\lambda)\epsilon') \leq \zeta_1(\lambda\epsilon_1 + (1-\lambda)\epsilon_1') + \zeta_2(\lambda\epsilon_2 + (1-\lambda)\epsilon_2')$. Applying the convexity of $\zeta_1$ and $\zeta_2$ to the right-hand side yields

$$\zeta(\lambda\epsilon + (1-\lambda)\epsilon') \leq \lambda(\zeta_1(\epsilon_1) + \zeta_2(\epsilon_2)) + (1-\lambda)(\zeta_1(\epsilon_1') + \zeta_2(\epsilon_2')).$$

Since this inequality holds for values that can be made arbitrarily close to $\lambda\zeta(\epsilon) + (1-\lambda)\zeta(\epsilon')$, the result follows.

3. Let $\epsilon \geq 0$ and $(\epsilon_1, \epsilon_2)$ be a pair such that $\epsilon_1 + \epsilon_2 = \epsilon$ and $\zeta_1(\epsilon_1) + \zeta_2(\epsilon_2)$ is arbitrarily close to $\zeta(\epsilon)$. Now for $\epsilon' \geq \epsilon$, consider the pair $(\epsilon_1, \epsilon_2 + (\epsilon' - \epsilon))$. The sum of its components is $\epsilon_1 + \epsilon_2 + \epsilon' - \epsilon = \epsilon'$. By the definition of $\zeta$, we have $\zeta(\epsilon') \leq \zeta_1(\epsilon_1) + \zeta_2(\epsilon_2 + (\epsilon' - \epsilon))$. Since $\zeta_2$ is non-increasing, it follows that $\zeta_2(\epsilon_2 + (\epsilon' - \epsilon)) \leq \zeta_2(\epsilon_2)$. This leads to the inequality $\zeta(\epsilon') \leq \zeta_1(\epsilon_1) + \zeta_2(\epsilon_2)$. As this holds for a value arbitrarily close to $\zeta(\epsilon)$, we conclude that $\zeta(\epsilon') \leq \zeta(\epsilon)$ as required.

4. This holds by definition. For $\epsilon = 0$, the only pair $(\epsilon_1, \epsilon_2)$ satisfying the constraints is $(0,0)$ so the infimum is taken over a single point.

5. We need to show that $\lim_{\epsilon\to 0^+}\zeta(\epsilon) = \zeta(0)$. From the fact that $\zeta$ is non-increasing, we already have $\zeta(\epsilon) \leq \zeta(0)$ for any $\epsilon \geq 0$. For the reverse inequality, consider an arbitrary $\delta > 0$. By the continuity of $\zeta_1$ and $\zeta_2$ at zero, there exists an $\eta > 0$ such that for any $x \in [0, \eta)$, both $\zeta_1(x) > \zeta_1(0) - \delta/2$ and $\zeta_2(x) > \zeta_2(0) - \delta/2$. Now, if we choose $\epsilon \in (0, \eta)$, then for any decomposition $\epsilon = \epsilon_1 + \epsilon_2$, both $\epsilon_1$ and $\epsilon_2$ must be less than $\eta$. This implies that any term in the infimum, $\zeta_1(\epsilon_1) + \zeta_2(\epsilon_2)$, is strictly greater than $\zeta_1(0) + \zeta_2(0) - \delta$. Therefore, the infimum itself must satisfy $\zeta(\epsilon) \geq \zeta(0) - \delta$, which completes the proof.

$\square$

We now use Lemmas 14, 15 and 16 and Corollary 13 to lower bound the optimality gap of disagreement discrepancy in terms of the optimality gap of the surrogates, evaluated on a subset of the input space.

**Theorem 4.** *Consider a classification task with $K > 2$ classes, where $h: \mathcal{X} \to [\![K]\!]^2$ is a reference model outputting a pair of class labels and $f: \mathcal{X} \to \mathbb{R}^K$ is a critic model outputting logits. Let $S, T$ be distributions on $\mathcal{X}$ and $\alpha > 0$. For $\lambda \in (0, 1)$ and $\delta \in (0, \frac{1-\lambda}{2})$, define a restricted input space:*

$$\mathcal{X}' = \left\{ x \in \mathcal{X} : h_1(x) = h_2(x), p_T(x) > 0, \lambda + \delta \leq \frac{p_S(x)}{\alpha p_T(x)} \leq 1 - \delta \right\}, \tag{8}$$

*Let $\hat{\mathrm{d}}_\alpha$ be either Rosenfeld & Garg's surrogate[5] with $\ell_{\mathrm{dis}} = \ell_{\mathrm{dis}}^{\mathrm{RG}}$, $\lambda = K/(2K-2)$, or Ginsberg et al.'s surrogate with $\ell_{\mathrm{dis}} = \ell_{\mathrm{dis}}^{\mathrm{GLK}}$, $\lambda = 1/(K-1)$, and $\ell_{\mathrm{agr}} = \ell_{\mathrm{ce}}$ in both cases. Then for both surrogates, there exists a convex function $\zeta : [0, \infty) \to [0, \infty)$ that is continuous at $0$ with $\zeta(0) = \delta/(1-\delta)\mathbf{1}_{S(\mathcal{X}')>0} + \alpha\delta\mathbf{1}_{T(\mathcal{X}')>0}$, such that*

$$\sup_{f' \in \mathcal{H}} \mathrm{d}_\alpha[h, f'](S, T) - \mathrm{d}_\alpha[h, f](S, T) \geq \zeta\left( \hat{\mathrm{d}}_\alpha[h, f](S|_{\mathcal{X}'}, T|_{\mathcal{X}'}) - \inf_{f' \in \mathcal{H}} \hat{\mathrm{d}}_\alpha[h, f'](S|_{\mathcal{X}'}, T|_{\mathcal{X}'}) \right).$$

*Proof.* For $i \in \{1, 2\}$, let $\ell_i = L_i[-\ell_{\mathrm{zo}}, \alpha\,\ell_{\mathrm{zo}}]$ and $\hat{\ell}_i = L_i[\ell_{\mathrm{ce}}, \alpha\,\ell_{\mathrm{dis}}]$. Using the fact that $\mathrm{d}_\alpha[h, f](S, T) = -\alpha\,\mathrm{d}_{\alpha^{-1}}[h, f](T, S)$ and expanding out the definitions, we have

$$\sup_{f' \in \mathcal{H}} \mathrm{d}_\alpha[h, f'](S, T) - \mathrm{d}_\alpha[h, f](S, T)$$

$$= \alpha\,\mathrm{d}_{\alpha^{-1}}[h, f](T, S) - \inf_{f' \in \mathcal{H}} \alpha\,\mathrm{d}_{\alpha^{-1}}[h, f'](T, S)$$

$$= R[\ell_1, h, \mathsf{A}f](S) - \inf_{f' \in \mathcal{H}} R[\ell_1, h, \mathsf{A}f'](S) + R[\ell_2, h, \mathsf{A}f](T) - \inf_{f' \in \mathcal{H}} R[\ell_2, h_2, \mathsf{A}f'](T)$$

$$= R[\Delta\,\ell_1, h, \mathsf{A}f](S) + R[\Delta\,\ell_2, h, \mathsf{A}f](T) \tag{16}$$

$$\geq R[\Delta\,\ell_1, h, \mathsf{A}f](S|_{\mathcal{X}'}) + R[\Delta\,\ell_2, h, \mathsf{A}f](T|_{\mathcal{X}'}) \tag{17}$$

Note that (16) follows since $f'$ can be optimized pointwise and (17) follows by replacing $R[\Delta\,\ell_1, h, \mathsf{A}f](S|_{\mathcal{X} \setminus \mathcal{X}'})$ and $R[\Delta\,\ell_2, h, \mathsf{A}f](T|_{\mathcal{X} \setminus \mathcal{X}'})$ by a lower bound of zero.

Next, we apply Corollary 13 to (17) on the restricted input space $\mathcal{X}'$ using $\ell_i(x, y, y')$ as the true loss and $\hat{\ell}_i(x, y, \mathbf{s})$ as the surrogate loss. Lemmas 14 and 15 ensure that $\Delta G[\ell_i, \hat{\ell}_i](\epsilon) = \delta/(1-\delta)\mathbf{1}_{i=1} + \alpha\delta\mathbf{1}_{i=2} > 0$ within a neighborhood of $\epsilon = 0$ for Rosenfeld & Garg and Ginsberg et al.'s surrogates respectively. This is needed to apply Corollary 13.

As a result, there exists a convex function $\zeta_1 : [0, \infty) \to [0, \infty)$ that is continuous at zero, with $\zeta_1(0) = \delta/(1-\delta)\mathbf{1}_{S(\mathcal{X}')>0}$ such that

$$R[\Delta\,\ell_1, h, \mathsf{A}f](S|_{\mathcal{X}'}) \geq \zeta_1\left( R[\Delta\,\hat{\ell}_1, h, f](S|_{\mathcal{X}'}) \right)$$

$$= \zeta_1\left( R[\hat{\ell}_1, h, f](S|_{\mathcal{X}'}) - \inf_{f' \in \mathcal{H}} R[\hat{\ell}_1, h, f'](S|_{\mathcal{X}'}) \right). \tag{18}$$

The last line follows since $f'$ can be optimized pointwise. By the same argument, there exists a convex function $\zeta_2 : [0, \infty) \to [0, \infty)$ that is continuous at zero with $\zeta_2(0) = \alpha\delta\mathbf{1}_{T(\mathcal{X}')>0}$ such that

$$R[\Delta\,\ell_2, h, \mathsf{A}f](T|_{\mathcal{X}'}) \geq \zeta_2\left( R[\hat{\ell}_2, h, f](T|_{\mathcal{X}'}) - \inf_{f' \in \mathcal{H}} R[\hat{\ell}_2, h, f'](T|_{\mathcal{X}'}) \right). \tag{19}$$

Now let

$$\zeta(\epsilon) = \inf_{\epsilon_1 + \epsilon_2 = \epsilon, \epsilon_1 \geq 0, \epsilon_2 \geq 0} \zeta_1(\epsilon_1) + \zeta_2(\epsilon_2). \tag{20}$$

---

[5]We consider a generalization of Rosenfeld & Garg's surrogate with $\alpha > 0$ and distinct reference models.

By Lemma 16, $\zeta$ is convex and continuous at zero, with $\zeta(0) = \zeta_1(0) + \zeta_2(0) = \delta/(1-\delta)\mathbf{1}_{S(\mathcal{X}')>0} + \alpha\delta\mathbf{1}_{T(\mathcal{X}')>0}$. Using the property that $\zeta(\epsilon_1 + \epsilon_2) \leq \zeta_1(\epsilon_1) + \zeta_2(\epsilon_2)$ from Lemma 16, along with (18), (19) and (17) yields

$$\sup_{f'\in\mathcal{H}} \mathrm{d}_\alpha[h, f'](S, T) - \mathrm{d}_\alpha[h, f](S, T)$$

$$\geq \zeta\left(R[\hat{\ell}_1, h, f](S|_{\mathcal{X}'}) - \inf_{f'\in\mathcal{H}} R[\hat{\ell}_1, h, f'](S|_{\mathcal{X}'}) + R[\hat{\ell}_2, h, f](T|_{\mathcal{X}'}) - \inf_{f'\in\mathcal{H}} R[\hat{\ell}_2, h, f'](T|_{\mathcal{X}'})\right)$$

$$= \zeta\left(\hat{\mathrm{d}}_\alpha[h, f](S|_{\mathcal{X}'}, T|_{\mathcal{X}'}) - \inf_{f'\in\mathcal{H}} \hat{\mathrm{d}}_\alpha[h, f'](S|_{\mathcal{X}'}, T|_{\mathcal{X}'})\right)$$

as required. $\qquad\square$

**Corollary 5.** *In the setting of Theorem 4, the surrogates proposed by Rosenfeld & Garg (2023) and Ginsberg et al. (2023) are not Bayes consistent for disagreement discrepancy when $K > 2$.*

*Proof.* We reuse definitions from the statement of Theorem 4. Using (7) and the fact that the critic $f'$ can be optimized pointwise, we can rewrite the surrogate optimality gap as

$$\hat{\mathrm{d}}_\alpha[h, f_n](S, T) - \inf_{f'\in\mathcal{H}} \hat{\mathrm{d}}_\alpha[h, f'](S, T) = R[\Delta\,\hat{\ell}_1, h, f_n](S) + R[\Delta\,\hat{\ell}_2, h, f_n](T)$$

$$\geq R[\Delta\,\hat{\ell}_1, h, f_n](S|_{\mathcal{X}'}) + R[\Delta\,\hat{\ell}_2, h, f_n](T|_{\mathcal{X}'})$$

$$= \hat{\mathrm{d}}_\alpha[h, f_n](S|_{\mathcal{X}'}, T|_{\mathcal{X}'}) - \inf_{f'\in\mathcal{H}} \hat{\mathrm{d}}_\alpha[h, f'](S|_{\mathcal{X}'}, T|_{\mathcal{X}'})$$

where the inequality follows by replacing $R[\Delta\,\hat{\ell}_1, h, \mathsf{A}f](S|_{\mathcal{X}\setminus\mathcal{X}'})$ and $R[\Delta\,\hat{\ell}_2, h, \mathsf{A}f](T|_{\mathcal{X}\setminus\mathcal{X}'})$ by a lower bound of zero. By the sandwich theorem, we have

$$\hat{\mathrm{d}}_\alpha[h, f_n](S, T) - \inf_{f'\in\mathcal{H}} \hat{\mathrm{d}}_\alpha[h, f'](S, T) \xrightarrow{p} 0$$

$$\implies \hat{\mathrm{d}}_\alpha[h, f_n](S|_{\mathcal{X}'}, T|_{\mathcal{X}'}) - \inf_{f'\in\mathcal{H}} \hat{\mathrm{d}}_\alpha[h, f'](S|_{\mathcal{X}'}, T|_{\mathcal{X}'}) \xrightarrow{p} 0.$$

Consider $\zeta$ as defined in (20) and let

$$X_n = \zeta\left(\hat{\mathrm{d}}_\alpha[h, f_n](S|_{\mathcal{X}'}, T|_{\mathcal{X}'}) - \inf_{f'\in\mathcal{H}} \hat{\mathrm{d}}_\alpha[h, f'](S|_{\mathcal{X}'}, T|_{\mathcal{X}'})\right).$$

Since $\zeta$ is continuous at zero, the continuous mapping theorem implies $X_n \xrightarrow{p} \zeta(0)$. Let $S, T$ be such that $S(\mathcal{X}') > 0$ or $T(\mathcal{X}') > 0$ where $\mathcal{X}'$ is defined in (8). Then $\zeta(0) = \delta/(1-\delta)\mathbf{1}_{S(\mathcal{X}')>0} + \alpha\delta\mathbf{1}_{T(\mathcal{X}')>0}$ is strictly positive.

We prove that

$$Y_n = \sup_{f'\in\mathcal{H}} \mathrm{d}_\alpha[h, f'](S, T) - \mathrm{d}_\alpha[h, f_n](S, T)$$

*does not* converge in probability to zero by contradiction. Suppose $Y_n \xrightarrow{p} 0$. Then by Slutsky's theorem, $X_n - Y_n \xrightarrow{p} \zeta(0)$. Now by preservation of inequality in the limit, if $Z_n \xrightarrow{p} Z$ and $Z_n \leq 0$ for all $n$, then $Z \leq 0$. Applying this to $Z_n = X_n - Y_n$ implies $\zeta(0) \leq 0$, which is a contradiction. $\qquad\square$

## D  PROOFS FOR SECTION 4.4

This appendix contains proofs for the results presented in Section 4.3. The key result of this section is Theorem 4, which we will prove using an upper bound of Zhang (2004b) that is analogous to our lower bound, presented in Corollary 13.

We begin by introducing a functional $\Delta H$ that relates the the true and surrogate excess losses. This functional plays a similar role as $\Delta G$ in Definition 9.

**Definition 17.** Let $\Delta H[\ell, \hat{\ell}]\colon [0, \infty) \times \mathcal{X} \times \mathcal{Y} \to [0, \infty)$ be a function such that for any $x \in \mathcal{X}$, $y \in \mathcal{Y}$,

$$\Delta H[\ell, \hat{\ell}](\epsilon, x, y) = \begin{cases} +\infty, & C[\ell](\epsilon, x, y) = \emptyset, \\ \inf_{z \in C[\ell](\epsilon, x, y)} \Delta \hat{\ell}(x, y, z), & \text{otherwise}, \end{cases}$$

where $C[\ell](\epsilon, x, y) = \{z \in \mathcal{Z} : \Delta \ell(x, y, \mathsf{A}z) \geq \epsilon\}$. We also define $\Delta H[\ell, \hat{\ell}]\colon [0, \infty) \to [0, \infty)$ such that

$$\Delta H[\ell, \hat{\ell}](\epsilon) = \inf_{(x,y) \in \mathcal{X} \times \mathcal{Y}} \Delta H[\ell, \hat{\ell}](\epsilon, x, y).$$

Having defined $\Delta H$, we are ready to restate the upper bound of Zhang (2004b) (labeled Corollary 26 in their paper). It provides an upper bound on the optimality gap of the true risk in terms of the optimality gap of the surrogate risk.

**Theorem 18** (Zhang, 2004b)**.** *If the loss function $\ell$ is bounded, and the function in Definition 17 satisfies $\forall \epsilon > 0$, $\Delta H[\ell, \hat{\ell}](\epsilon) > 0$, then there exists a concave function $\xi\colon [0, \infty) \to [0, \infty)$ that depends only on $\ell$, $\hat{\ell}$, such that $\xi(0) = 0$ and $\lim_{\delta \to 0^+} \xi(\delta) = 0$. Moreover, given $h\colon \mathcal{X} \to \mathcal{Y}$, $f\colon \mathcal{X} \to \mathcal{Z}$ we have*

$$\mathop{\mathrm{E}}_{X \sim D} \Delta \ell(X, h(X), f(X)) \leq \xi\left(\mathop{\mathrm{E}}_{X \sim D} \Delta \hat{\ell}(X, h(X), f(X))\right)$$

*for all distributions $D$ on $\mathcal{X}$.*

Before we can apply this result to prove Theorem 6, we must prove that the positivity condition on $\Delta H$ holds for the true/surrogate losses of interest. We do this in the lemma below.

**Lemma 19.** *Consider a classification setting where the reference model output space consists of pairs of labels $\mathcal{Y} = [\![K]\!]^2$, and the evaluation model output space consists of logits $\mathcal{Z} = \mathbb{R}^K$. Let $\mathsf{A}\colon \mathcal{X} \to [\![K]\!]$ be as defined in (1). Consider the true loss $\ell_i = L_i[\ell_{\mathrm{zo}}, -\ell_{\mathrm{zo}}]$ and corresponding surrogate loss $\hat{\ell}_i = L_i[\ell_{\mathrm{ce}}, \ell_{\mathrm{dis}}^{\mathrm{Ours}}]$, where $L_i$ is defined in (5), $\ell_{\mathrm{ce}}$ is defined in (2), and $\ell_{\mathrm{dis}}^{\mathrm{Ours}}$ is defined in (9). Then for $i \in \{1, 2\}$ and all $\epsilon > 0$ we have $\Delta H[\ell_i, \hat{\ell}_i](\epsilon) > 0$.*

*Proof.* We prove the result by evaluating $\Delta H[\ell_i, \hat{\ell}_i](\epsilon, x, \mathbf{y})$ directly for four cases: $i = 1$ and $y_1 = y_2$, $i = 2$ and $y_1 \neq y_2$, $i = 2$ and $y_1 = y_2$, and $i = 2$ and $y_1 \neq y_2$. For brevity, we define $\mathbf{q} \coloneqq [\mathrm{e}^{s_1}, \ldots, \mathrm{e}^{s_K}] / \sum_c \mathrm{e}^{s_c}$ and $r(x) \coloneqq \alpha p_T(x) / p_S(x)$.

**Case 1:** $i = 1$ and $y_1 = y_2$. Using the expression for $\Delta \ell_1$ derived in Appendix B.1 we have

$$C[\ell_1](\epsilon, x, \mathbf{y}) = \left\{\mathbf{s} \in \mathbb{R}^K : \mathbf{1}_{r(x) \leq \alpha}(\mathbf{1}_{s_{y_1} < \max_{c \neq y_1} s_c} - \mathbf{1}_{r(x) > 1})(1 - r(x)) \geq \epsilon\right\}.$$

For $r(x) \leq 1$, this set is non-empty when $s_{y_1} < \max_{c \neq y_1} s_c$, $r(x) \leq 1 - \epsilon$ and $r(x) \leq \alpha$. Using the expression for $\Delta \hat{\ell}_1$ derived in Appendix B.2, we have

$$\begin{aligned} \Delta H[\ell_1, \hat{\ell}_1](\epsilon, x, \mathbf{y}) &= \inf_{\mathbf{s} \in C[\ell_1](\epsilon, x)} \Delta \hat{\ell}_1(x, \mathbf{y}, \mathbf{s}) \\ &= \inf_{q_{y_1} < \frac{1}{2}} -\log(q_{y_1}(1 + r(x))) - r(x) \log\left((1 - q_{y_1})(1 + r(x)^{-1})\right) \\ &= \log\left(\frac{2}{1 + r(x)}\right) + r(x) \log\left(\frac{2}{1 + r(x)^{-1}}\right) \end{aligned}$$

Similarly, when $r(x) > 1$ the set $C[\ell_1](\epsilon, x, \mathbf{y})$ is non-empty when $q_{y_1} = \max_c q_c$, $r(x) \geq \epsilon + 1$ and $r(x) \leq \alpha$, where we have

$$\begin{aligned} \Delta H[\ell_1, \hat{\ell}_1](\epsilon, x, \mathbf{y}) &= \inf_{\mathbf{s} \in C[\ell_1](\epsilon, x)} \Delta \hat{\ell}_1(x, \mathbf{y}, \mathbf{s}) \\ &= \inf_{q_{y_1} > \frac{1}{2}} -\log(q_{y_1}(1 + r(x))) - r(x) \log\left((1 - q_{y_1})(1 + r(x)^{-1})\right) \\ &= \log\left(\frac{2}{1 + r(x)}\right) + r(x) \log\left(\frac{2}{1 + r(x)^{-1}}\right). \end{aligned}$$

Thus we have

$$\Delta H[\ell_1, \hat{\ell}_1](\epsilon, x, \mathbf{y}) \geq \min_{r(x) \in \{\min\{1-\epsilon, \alpha\}, 1+\epsilon\}} \log\left(\frac{2}{1+r(x)}\right) + r(x) \log\left(\frac{2}{1+r(x)^{-1}}\right) > 0.$$

**Case 2:** $i = 1$ with $y_1 \neq y_2$. Using the expression for $\Delta \ell_1$ we have

$$C[\ell_1](\epsilon, x, \mathbf{y}) = \left\{\mathbf{s} \in \mathbb{R}^K : \mathbf{1}_{r(x) \leq \alpha}\left(\mathbf{1}_{s_{y_1} < \max_{c \neq y_1} s_c} + r(x)\mathbf{1}_{s_{y_2} = \max_c s_c}\right) \geq \epsilon\right\}.$$

This set can be partitioned into subsets:

$$C_1[\ell_1](\epsilon, x, \mathbf{y}) = \left\{\mathbf{s} \in \mathbb{R}^K : s_{y_1} < \max_{c \neq y_1} s_c, s_{y_2} \neq \max_c s_c\right\}, \quad \text{for } r(x) \leq \alpha, \epsilon \leq 1,$$

$$C_2[\ell_1](\epsilon, x, \mathbf{y}) = \left\{\mathbf{s} \in \mathbb{R}^K : s_{y_1} < \max_{c \neq y_1} s_c, s_{y_2} = \max_c s_c\right\}, \quad \text{for } \max\{0, \epsilon-1\} \leq r(x) \leq \alpha,$$

where $C_1[\ell_2](\epsilon, x, \mathbf{y}) = C_2[\ell_2](\epsilon, x, \mathbf{y}) = \emptyset$ outside the specified domains.

For $r(x) \leq \alpha, \epsilon \leq 1$ we have

$$\inf_{\mathbf{s} \in C_1[\ell_1](\epsilon, x, \mathbf{y})} \Delta \hat{\ell}_1(x, \mathbf{y}, \mathbf{s}) = \inf_{\mathbf{s} \in C_1[\ell_1](\epsilon, x, \mathbf{y})} -\log(q_{y_1}) - r(x)\log(1 - q_{y_2})$$

$$= \log 2,$$

and for $\max\{0, \epsilon-1\} \leq r(x) \leq \alpha$ we have

$$\inf_{\mathbf{s} \in C_1[\ell_1](\epsilon, x, \mathbf{y})} \Delta \hat{\ell}_1(x, \mathbf{y}, \mathbf{s}) = \inf_{\mathbf{s} \in C_1[\ell_1](\epsilon, x, \mathbf{y})} -\log(q_{y_1}) - r(x)\log(1 - q_{y_2})$$

$$= \lim_{\eta \to 0^+} \log(K + \eta) + r(x)\log(K - \eta),$$

$$\geq \log K.$$

Thus we have

$$\Delta H[\ell_1, \hat{\ell}_1](\epsilon, x, \mathbf{y}) \geq \min\{\log 2, \log K\} > 0.$$

**Case 3:** $i = 2$ with $y_1 = y_2$. Using the expression for $\Delta \ell_2$ derived in Appendix B.1, we have

$$C[\ell_2](\epsilon, x, \mathbf{y}) = \left\{\mathbf{s} \in \mathbb{R}^K : \alpha\mathbf{1}_{r(x)^{-1} < \alpha^{-1}}(\mathbf{1}_{s_{y_1} < \max_{c \neq y_1} s_c} - \mathbf{1}_{r(x)^{-1} < 1})(r(x)^{-1} - 1) \geq \epsilon\right\}.$$

For $r(x)^{-1} \geq 1$ this set is non-empty when $s_{y_1} < \max_{c \neq y_1} s_c$, $r(x)^{-1} \geq 1 + \epsilon/\alpha$ and $r(x)^{-1} < 1/\alpha$. Using the expression for $\Delta \hat{\ell}_2$ derived in Appendix B.2, we have

$$\Delta H[\ell_2, \hat{\ell}_2](\epsilon, x, \mathbf{y}) = \inf_{\mathbf{s} \in C[\ell_2](\epsilon, x, \mathbf{y})} \Delta \hat{\ell}_2(x, \mathbf{y}, \mathbf{s})$$

$$= \alpha \inf_{q_{y_1} < \frac{1}{2}} -r(x)^{-1}\log(q_{y_1}(1 + r(x))) - \log((1 - q_{y_1})(1 + r(x)^{-1}))$$

$$= \alpha r(x)^{-1}\log\left(\frac{2}{1+r(x)}\right) + \alpha\log\left(\frac{2}{1+r(x)^{-1}}\right)$$

On the other hand, when $r(x)^{-1} < 1$ the set $C[\ell_2](\epsilon, x, \mathbf{y})$ is non-empty when $s_{y_1} = \max_c s_c$, $r(x)^{-1} \leq 1 - \epsilon/\alpha$ and $r(x)^{-1} < 1/\alpha$, where we have

$$\Delta H[\ell_2, \hat{\ell}_2](\epsilon, x, \mathbf{y}) = \inf_{\mathbf{s} \in C[\ell_2](\epsilon, x, \mathbf{y})} \Delta \hat{\ell}_2(x, \mathbf{y}, \mathbf{s})$$

$$= \alpha \inf_{q_{y_1} > \frac{1}{2}} -r(x)^{-1}\log(q_{y_1}(1 + r(x))) - \log((1 - q_{y_1})(1 + r(x)^{-1}))$$

$$= \alpha r(x)^{-1}\log\left(\frac{2}{1+r(x)}\right) + \alpha\log\left(\frac{2}{1+r(x)^{-1}}\right).$$

Thus we have

$$\Delta H[\ell_2, \hat{\ell}_2](\epsilon, x, \mathbf{y}) \geq \min_{r(x)^{-1} \in \{\min\{\frac{1}{\alpha}, 1-\frac{\epsilon}{\alpha}\}, 1+\frac{\epsilon}{\alpha}\}} \alpha r(x)^{-1}\log\left(\frac{2}{1+r(x)}\right) + \alpha\log\left(\frac{2}{1+r(x)^{-1}}\right)$$

$$> 0.$$

**Case 4:** $i = 2$ and $y_1 \neq y_2$. Using the expression for $\Delta \ell_2$ we have

$$C[\ell_2](\epsilon, x, \mathbf{y}) = \left\{ \mathbf{s} \in \mathbb{R}^K : \alpha \mathbf{1}_{r(x)^{-1} < \alpha^{-1}} \left( r(x)^{-1} \mathbf{1}_{s_{y_1} < \max_{c \neq y_1} s_c} + \mathbf{1}_{s_{y_2} = \max_c s_c} \right) \geq \epsilon \right\}.$$

This set can be partitioned into subsets:

$$C_1[\ell_2](\epsilon, x, \mathbf{y}) = \left\{ \mathbf{s} \in \mathbb{R}^K : s_{y_1} < \max_{c \neq y_1} s_c, s_{y_2} \neq \max_c s_c \right\}, \quad \text{for } \frac{\epsilon}{\alpha} \leq r(x)^{-1} < \alpha,$$

$$C_2[\ell_2](\epsilon, x, \mathbf{y}) = \left\{ \mathbf{s} \in \mathbb{R}^K : s_{y_1} < \max_{c \neq y_1} s_c, s_{y_2} = \max_c s_c \right\}, \quad \text{for } \max \left\{ 0, \frac{\epsilon}{\alpha} - 1 \right\} \leq r(x)^{-1} < \alpha,$$

where $C_1[\ell_2](\epsilon, x, \mathbf{y}) = C_2[\ell_2](\epsilon, x, \mathbf{y}) = \emptyset$ outside the specified domains.

For $\frac{\epsilon}{\alpha} \leq r(x)^{-1} < \alpha$, we have

$$\inf_{\mathbf{s} \in C_1[\ell_2](\epsilon, x, \mathbf{y})} \Delta \hat{\ell}_2(x, \mathbf{y}, \mathbf{s}) = \inf_{\mathbf{s} \in C_1[\ell_2](\epsilon, x, \mathbf{y})} \alpha r(x)^{-1} \log(q_{y_1}) + \alpha \log(1 - q_{y_2})$$

$$= \alpha r(x)^{-1} \log 2$$

$$\geq \epsilon \log 2,$$

and for $\max\{0, \frac{\epsilon}{\alpha} - 1\} \leq r(x)^{-1} < \alpha$ we have

$$\inf_{\mathbf{s} \in C_2[\ell_2](\epsilon, x, \mathbf{y})} \Delta \hat{\ell}_2(x, \mathbf{y}, \mathbf{s}) = \inf_{\mathbf{s} \in C_2[\ell_2](\epsilon, x, \mathbf{y})} \alpha r(x)^{-1} \log(q_{y_1}) + \alpha \log(1 - q_{y_2})$$

$$= \lim_{\eta \to 0^+} \alpha r(x)^{-1} \log(K + \eta) + \alpha \log(K - \eta)$$

$$\geq \alpha \log K.$$

Thus we have

$$\Delta H[\ell_2, \hat{\ell}_2](\epsilon, x, \mathbf{y}) \geq \min\{\epsilon \log 2, \alpha \log K\} > 0.$$

Hence we have shown that $\Delta H[\ell_i, \hat{\ell}_i](\epsilon, x, \mathbf{y}) > 0$ for any $x \in \mathcal{X}$, $\mathbf{y} \in [\![K]\!]^2$ and $\epsilon > 0$, which implies $\Delta H[\ell_i, \hat{\ell}_i](\epsilon) > 0$ as required. $\qquad \square$

We also need the following result which allows us to compose the convex functions that appear in the framework of Zhang (2004b, Appendix A).

**Lemma 20.** *Let* $\xi_1, \xi_2 \colon [0, \infty) \to [0, \infty)$ *be convex functions that are continuous at zero and non-increasing. Then the function* $\xi \colon [0, \infty) \to [0, \infty)$ *such that*

$$\xi(\epsilon) = \sup_{\epsilon_1 + \epsilon_2 = \epsilon, \epsilon_1 \geq 0, \epsilon_2 \geq 0} \xi_1(\epsilon_1) + \xi_2(\epsilon_2)$$

*has the following properties:*

1. *it satisfies* $\xi_1(\epsilon_1) + \xi_2(\epsilon_2) \leq \xi(\epsilon_1 + \epsilon_2)$ *for any* $\epsilon_1, \epsilon_2 \in [0, \infty)$,

2. *it is concave,*

3. *it is non-decreasing,*

4. *it satisfies* $\xi(0) = \xi_1(0) + \xi_2(0)$, *and*

5. *it is continuous at zero.*

*Proof.* The proof follows a similar structure as the proof of Lemma 16. $\qquad \square$

We now use Lemma 19 and Theorem 18 to upper bound the optimality gap of disagreement discrepancy in terms of the optimality gap of our surrogate.

**Theorem 6.** *Consider a classification task where $h\colon \mathcal{X} \to [\![K]\!]^2$ is a reference model outputting a pair of class labels and $f\colon \mathcal{X} \to \mathbb{R}^K$ is a critic model outputting logits. For any $\alpha > 0$, let $\hat{\mathrm{d}}_\alpha$ be our surrogate with $\ell_{\mathrm{dis}} = \ell_{\mathrm{dis}}^{\mathrm{Ours}}$ and $\ell_{\mathrm{agr}} = \ell_{\mathrm{ce}}$. Then, for any distributions $S, T$ on $\mathcal{X}$, there exists a concave function $\xi \colon [0, \infty) \to [0, \infty)$ that is continuous at $0$ with $\xi(0) = 0$, such that*

$$\sup_{f \in \mathcal{H}} \mathrm{d}_\alpha[h, f'](S, T) - \mathrm{d}_\alpha[h, f](S, T) \leq \xi\left( \hat{\mathrm{d}}_\alpha[h, f](S, T) - \inf_{f' \in \mathcal{H}} \hat{\mathrm{d}}_\alpha[h, f'](S, T) \right).$$

*Proof.* Using the fact that $\mathrm{d}_\alpha[h, f](S, T) = -\alpha \, \mathrm{d}_{\alpha^{-1}}[(h_2, h_1), f](T, S)$ and expanding out the definitions, we have

$$\sup_{f' \in \mathcal{H}} \mathrm{d}_\alpha[h, f'](S, T) - \mathrm{d}_\alpha[h, f](S, T) = \alpha \, \mathrm{d}_{\alpha^{-1}}[(h_2, h_1), f](T, S)$$

$$- \alpha \inf_{f' \in \mathcal{H}} \mathrm{d}_{\alpha^{-1}}[(h_2, h_1), f'](T, S)$$

$$= R[\ell_1, h, \mathsf{A}f](S) - \inf_{f' \in \mathcal{H}} R[\ell_1, h, \mathsf{A}f'](S)$$

$$+ R[\ell_2, h, \mathsf{A}f](T) - \inf_{f' \in \mathcal{H}} R[\ell_2, h, \mathsf{A}f'](T)$$

$$= R[\Delta\,\ell_1, h, \mathsf{A}f](S) + R[\Delta\,\ell_2, h, \mathsf{A}f](T),$$

where the last equality follows since $f'$ can be optimized pointwise. Applying Theorem 18 to each of these terms gives

$$\sup_{f' \in \mathcal{H}} \mathrm{d}_\alpha[h, f'](S, T) - \mathrm{d}_\alpha[h, f](S, T) \leq \xi_1\left( R[\Delta\,\hat{\ell}_1, h, f](S) \right) + \xi_2\left( R[\Delta\,\hat{\ell}_2, h, f](T) \right), \quad (21)$$

where $\xi_1$ and $\xi_2$ are concave non-decreasing functions that are continuous at zero and satisfy $\xi_1(0) = \xi_2(0) = 0$. Note that in order to invoke Theorem 18, we have used Lemma 19 which guarantees positivity of $\Delta H[\ell_i, \hat{\ell}_i](\epsilon)$ for any $\epsilon > 0$. We have also used the fact that $\ell_1$ and $\ell_2$ are bounded.

Next, we define the function $\xi \colon [0, \infty) \to [0, \infty)$

$$\xi(\delta) = \sup_{\delta_1 + \delta_2 = \delta, \delta_1 \geq 0, \delta_2 \geq 0} \xi_1(\delta_1) + \xi_2(\delta_2),$$

which is concave, continuous at zero and satisfies $\xi(0) = \xi_1(0) + \xi_2(0) = 0$ by Lemma 20. Combining the first property of Lemma 20 with (21) yields

$$\sup_{f' \in \mathcal{H}} \mathrm{d}_\alpha[h, f'](S, T) - \mathrm{d}_\alpha[h, f](S, T) \leq \xi\left( R[\Delta\,\hat{\ell}_1, h, f](S) + R[\Delta\,\hat{\ell}_2, h, f](T) \right)$$

$$= \xi\left( \hat{\mathrm{d}}_\alpha[h, f](S, T) - \inf_{f' \in \mathcal{H}} \hat{\mathrm{d}}_\alpha[h, f'](S, T) \right),$$

where we have again used the fact that $f'$ can be optimized pointwise in the last equality. $\qquad \square$

**Corollary 7.** *Our surrogate for disagreement discrepancy with cross-entropy agreement loss and the disagreement loss specified in (9) is Bayes consistent for all $K \geq 2$.*

*Proof.* The result follows from Theorem 6. Let $\hat{G}_n = \hat{\mathrm{d}}_\alpha[h, f_n](S, T) - \inf_{f' \in \mathcal{H}} \hat{\mathrm{d}}_\alpha[h, f'](S, T)$ and $G_n = \sup_{f' \in \mathcal{H}} \mathrm{d}_\alpha[h, f'](S, T) - \mathrm{d}_\alpha[h, f_n](S, T)$. By the continuous mapping theorem and continuity of $\xi$ at zero, we have $\xi(\hat{G}_n) \xrightarrow{p} 0$. Also, by Theorem 6, we have $0 \leq G_n \leq \xi(\hat{G}_n)$ for all $n$. Applying the sandwich theorem yields the desired result: $G_n \xrightarrow{p} 0$. $\qquad \square$

## E    ROSENFELD AND GARG'S ERROR BOUND UNDER COVARIATE SHIFT

This appendix presents the error bound for models under covariate shift, as proposed by Rosenfeld & Garg (2023). While this content is not novel, we include it here to make our paper self-contained and to provide necessary context for our experiments in Section 5 and the attack described in Appendix F.

Rosenfeld & Garg (2023) developed a method to bound the error of a model under distribution shift using disagreement discrepancy, requiring only labeled source data and unlabeled target data. Their

approach involves training a critic model, chosen from a specified hypothesis class, to maximize the disagreement discrepancy between itself and the model under evaluation. This maximized disagreement discrepancy is then used to construct a probabilistic upper bound on the target error. The bound is formally stated in the following theorem:

**Theorem 21** (Error bound). *Let $S_{\mathrm{tr}}, S_{\mathrm{te}}$ and $T_{\mathrm{tr}}, T_{\mathrm{te}}$ be train and test datasets drawn i.i.d. from the source distribution $S$ and target distribution $T$, respectively. Let $y^\star \colon \mathcal{X} \to [\![K]\!]$ be the ground truth labeling function, $h \colon \mathcal{X} \to [\![K]\!]$ be the model under evaluation and $f^\star \in \arg\min_{f \in \mathcal{H}} \hat{\mathrm{d}}[h, f](S_{\mathrm{tr}}, T_{\mathrm{tr}})$ be the optimal critic within hypothesis class $\mathcal{H} \subseteq \{f \colon \mathcal{X} \to \mathbb{R}^K\}$. Assume $\mathrm{d}[h, y^\star](S, T) \leq \mathrm{d}[h, f^\star](S, T)$. Then with probability $1 - \delta$ we have*

$$\underbrace{R[\ell_{\mathrm{zo}}, y^\star, h](T)}_{\text{pop. error on target}} \leq \underbrace{R[\ell_{\mathrm{zo}}, y^\star, h](S_{\mathrm{te}})}_{\text{emp. error on source}} + \underbrace{\mathrm{d}[h, f^\star](S_{\mathrm{te}}, T_{\mathrm{te}})}_{\text{emp. disagreement discrepancy}} + \underbrace{\sqrt{\frac{(|S_{\mathrm{te}}| + 4|T_{\mathrm{te}}|) \log 1/\delta}{2|S_{\mathrm{te}}||T_{\mathrm{te}}|}}}_{\text{sample correction}}.$$

The accuracy of this bound critically depends on estimating the disagreement discrepancy term. Here, the consistency of the surrogate plays a crucial role. A consistent surrogate ensures that, asymptotically, optimizing it leads to the same result as optimizing the true disagreement discrepancy. This provides theoretical justification for using the surrogate in training the critic $f^\star$ and, consequently, in estimating the upper bound.

The bound's validity also relies on a key assumption: $\mathrm{d}[h, y^\star](S, T) \leq \mathrm{d}[h, f^\star](S, T)$. This assumption states that the disagreement discrepancy achieved by the optimal critic $f^\star$ should be at least as large as that achieved by the ground truth labeling function $y^\star$. The extent to which this assumption holds depends on several factors:

- The expressiveness of the critic's hypothesis class $\mathcal{H}$
- The quality of the surrogate disagreement discrepancy
- The effectiveness of the optimization procedure

When these factors align favorably, the trained critic $f^\star$ should achieve a disagreement discrepancy that meets or exceeds what would be obtained using the ground truth labeling function $y^\star$. However, as we observe in our experiments (Section 5), this assumption does not always hold in practice, leading to potential violations of the bound.

The analysis of surrogate consistency and its impact on the reliability of this bound forms the core of our work, as detailed in the main text.

## F    ATTACKING ROSENFELD & GARG'S ERROR BOUND

This appendix describes an attack on the error bound of Rosenfeld & Garg (2023), aiming to underestimate the error by perturbing the target data. We perturb a fraction of the inputs, constraining the perturbation for each input in $\ell_\infty$-norm to $\epsilon$. The attack can be viewed as an application of projected gradient descent.

Let

$$\mathrm{ub}(T_{\mathrm{tr}}, T_{\mathrm{te}}) = R[\ell_{\mathrm{zo}}, y^\star, h](S_{\mathrm{te}}) + \mathrm{d}[h, f^\star(T_{\mathrm{tr}})](S_{\mathrm{te}}, T_{\mathrm{te}}) + \sqrt{\frac{(|S_{\mathrm{te}}| + 4|T_{\mathrm{te}}|) \log 1/\delta}{2|S_{\mathrm{te}}||T_{\mathrm{te}}|}}$$

denote the upper bound on the target error from Theorem 21. Note that we've dropped the dependence on the source train/test datasets $S_{\mathrm{tr}}$, $S_{\mathrm{te}}$, and made explicit that the critic model $f^\star(T_{\mathrm{tr}}) \in \arg\max_{f \in \mathcal{H}} \hat{\mathrm{d}}[h, f](S_{\mathrm{tr}}, T_{\mathrm{tr}})$ depends on the target training dataset $T_{\mathrm{tr}}$.

Our objective as the attacker is to minimize the difference between the upper bound and the actual target error $\mathrm{ub}(T_{\mathrm{tr}}, T_{\mathrm{te}}) - R[\ell_{\mathrm{zo}}, y^\star, h](T_{\mathrm{te}})$, subject to the $\ell_\infty$ constraint on the perturbation to the target train and test instances. If this difference becomes negative, we have triggered a violation of the bound. By dropping terms that are constant with respect to the target datasets, we can equivalently minimize:

$$R[\ell_{\mathrm{zo}}, h, \mathsf{A}f^\star(T_{\mathrm{tr}})](T_{\mathrm{te}}) - R[\ell_{\mathrm{zo}}, y^\star, h](T_{\mathrm{te}}).$$

This problem cannot be solved using gradient-based optimization, as the zero-one loss $\ell_{\mathrm{zo}}$ is not differentiable. Additionally, $f^\star(T_{\mathrm{tr}})$, which is the solution to an optimization problem, is difficult to differentiate through.

To address these challenges, we employ a two-step optimization procedure that uses differentiable surrogate losses. For this attack scenario, we make an exception to our usual treatment of the reference model $h$. While throughout the paper we've assumed $h$ returns raw outputs (class labels), here we need access to its logits or class probabilities to compute gradients with respect to the target data. This is necessary for the attack but does not change our general framework.

Our two-step procedure is as follows:

1. Optimize the target test data $T_{\mathrm{te}}$:
   - Replace $R[\ell_{\mathrm{zo}}, h, \mathsf{A}f^\star(T_{\mathrm{tr}})](T_{\mathrm{te}})$ with $-R[\ell_{\mathrm{dis}}, h, f^\star(T_{\mathrm{tr}})](T_{\mathrm{te}})$, where $\ell_{\mathrm{dis}}(x, \mathbf{p}, \mathbf{s})$ takes as input a probability vector $\mathbf{p} \in \Lambda_{K-1}$ over $K$ classes instead of a class label. This encourages agreement between the reference model $h$ and critic model $f^\star$ (due to the minus sign). We use the Gumbel softmax trick to make this loss differentiable in the class probabilities output by $h$.
   - Replace $R[\ell_{\mathrm{zo}}, y^\star, h](T_{\mathrm{te}})$ with $-R[\ell_{\mathrm{ce}}, y^\star, h](T_{\mathrm{te}})$, encouraging disagreement between the reference model $h$ and ground truth labeling function $y^\star$ (due to the minus sign).
   - Compute the gradient of the surrogate with respect to the selected target test inputs, take a gradient descent step, and project the perturbation onto the $\ell_\infty$-norm ball of radius $\epsilon$ centered on the original target input.

2. Update the target training data $T_{\mathrm{tr}}$ using the same surrogate objective and algorithm as for the test data, followed by updating the model $f^\star(T_{\mathrm{tr}})$.

Importantly, since the objective consists of sums over target data, we can optimize the target inputs in batches. This allows us to attack large datasets without needing to load all inputs into memory simultaneously, which would be infeasible for datasets with tens of thousands of images.

## G  FURTHER EXPERIMENTAL DETAILS AND RESULTS

### G.1  REPLICATION OF EXPERIMENTS WITH EXISTING AND NEW SURROGATES

This appendix provides additional details and results for the replication of experiments from Rosenfeld & Garg (2023), complementing the results and discussion in Section 5.1.1.

**Experimental Setup**   We briefly describe key elements of the experimental setup in an effort to make our paper self-contained. For comprehensive details, readers are referred to Appendix A of Rosenfeld & Garg (2023) and their publicly released code[6]. The experiments utilize 11 publicly available vision datasets commonly used in distribution shift contexts:

- CIFAR10 (Krizhevsky & Hinton, 2009): Original as source; CIFAR10v2 (Recht et al., 2018) and CIFAR10-C (Hendrycks & Dietterich, 2019) as targets.
- CIFAR100 (Krizhevsky & Hinton, 2009): Original as source; CIFAR100-C (Hendrycks & Dietterich, 2019) as target.
- FMoW from WILDS (Koh et al., 2021): Train split as source; other splits (collected at later times) as targets.
- Camelyon17 from WILDS (Koh et al., 2021): Train split as source; other splits (from different hospitals) as targets.
- BREEDS (Santurkar et al., 2021): Datasets derived from ImageNet (Russakovsky et al., 2015), including entity13, entity30, living17, and nonliving26. Original ImageNet subpopulation 1 as source; subpopulation 2 and ImageNetv2 (Recht et al., 2019) subpopulations 1 and 2 as targets.

---

[6]https://github.com/erosenfeld/disagree_discrep

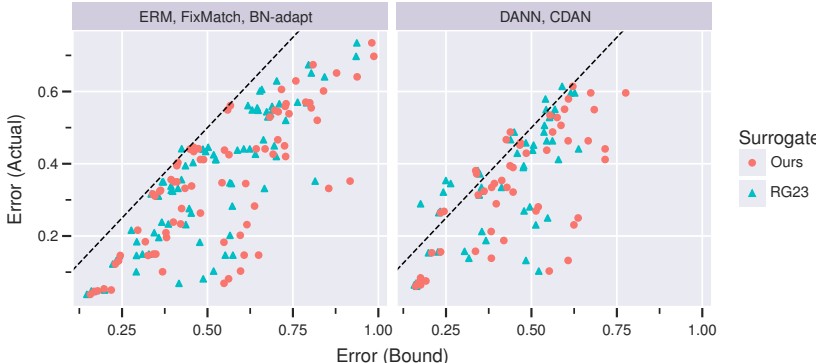

Figure 5: Comparison of error bounds versus actual error on target data for our surrogate and that of Rosenfeld & Garg (2023). Each point represents a shift/model, with points above the dashed line indicating bound violations. Results are disaggregated by training method: non-domain adversarial training (left) versus domain-adversarial training (right).

- OfficeHome (Venkateswara et al., 2017): Product domain as source, other domains as targets.

- DomainNet (Peng et al., 2019): Real domain as source, other domains as targets.

- Syn2Real (Peng et al., 2018): Train split (synthetic object renders) as source, other splits (real object images) as targets.

Models used include ResNet-18, ResNet-50 (He et al., 2016), and DenseNet-121 (Huang et al., 2017), depending on the dataset. Generally, models are pretrained on ImageNet and then fine-tuned on the source dataset. Training/fine-tuning on source data employs five methods: empirical risk minimization (ERM), FixMatch (Sohn et al., 2020), BN-adapt (Li et al., 2017), CDAN (Long et al., 2018), or DANN (Ganin et al., 2016), with data augmentation applied during training.

Critics are implemented as linear models that consume either logits or features from the model under evaluation. Specifically, the weights of the evaluated model are frozen, and only an appended linear layer is tunable. Unless otherwise specified, logit-based critics are used. All loss functions involving softmax operations, including the standard cross-entropy loss and our proposed disagreement loss, are implemented in log-space to ensure numerical stability.

**Additional Results**  In Section 5.1.1, we compared surrogates based on their resulting estimates for the disagreement discrepancy, where larger estimates are superior. This is the only term in Rosenfeld & Garg's error bound (Theorem 21) that depends on the surrogate. Here, we provide additional results comparing the complete error bound (including all terms) with the actual error on labeled target data.

Figure 5 compares the error bound and actual error for numerous shift/model pairs. We disaggregate by model training method: domain adversarial training (DANN, CDAN) on the right and non-domain adversarial training methods (ERM, FixMatch, BN-adapt) on the left. We observe more violations of the bound (points above the dashed line) for models trained using domain adversarial methods. Specifically, there are 13 violations when using Rosenfeld & Garg's surrogate versus 6 violations for our surrogate. Rosenfeld & Garg (2023) attribute this to DANN and CDAN penalizing the ability to discriminate between source and target distributions in feature space, effectively minimizing the disagreement discrepancy term in the error bound. They argue that this scenario violates the assumption of their bound, as DANN and CDAN can produce models that are, in some sense, worst-case (adversarial) for the bound.

We also consider a different critic architecture. Figure 6 compares logit-based and feature-based critic architectures. Rosenfeld & Garg (2023) suggest that feature-based critics tend to have greater capacity than logit-based critics, resulting in more conservative error bounds. Our results confirm this behavior for critics trained with our surrogate. Note that this figure only includes models trained using ERM, FixMatch, and BN-adapt, excluding domain adversarial trained models.

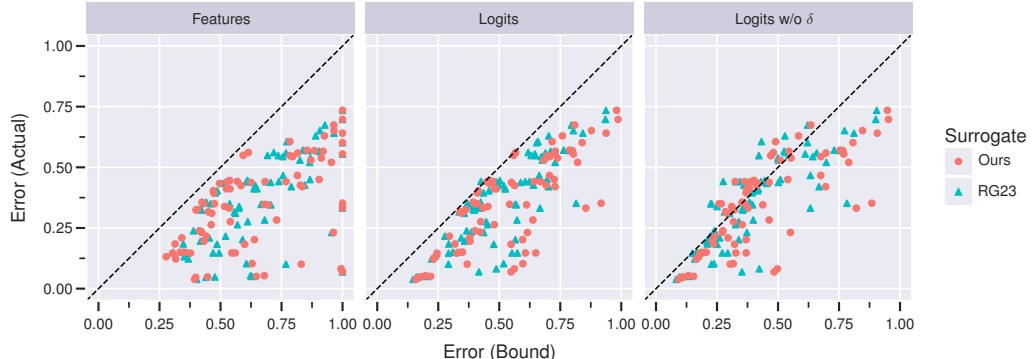

Figure 6: Comparison of error bounds versus actual error on target data for our surrogate and that of Rosenfeld & Garg (2023). Each point represents a shift/model, with points above the dashed line indicating bound violations. Only models trained using ERM, FixMatch, and BN-adapt are included. Results are disaggregated by critic architecture: feature-based (left), logit-based (middle), and logit-based without $\delta$ term (right).

## G.2 ROBUSTNESS TO ADVERSARIALLY CHOSEN DATA

This appendix provides additional results and details about the experiments assessing the robustness of Rosenfeld & Garg's error bounds to adversarially chosen target data, complementing the results presented in Section 5.1.2.

**Experimental Setup**    Due to the unavailability of the exact models and train/test splits used in the replication experiments, we attempted to recreate the setup as described in Appendix A of Rosenfeld & Garg (2023) and partially outlined in our Appendix G.1. We used 8 of the 11 datasets listed in Appendix G.1: CIFAR10, CIFAR100, FMoW, BREEDS (entity13, entity30, living17, nonliving26), and OfficeHome. DomainNet and Syn2Real were excluded due to their qualitative similarity to OfficeHome (shifts based on image style). Camelyon17 was omitted as it is a binary classification dataset where the surrogates for disagreement discrepancy are equivalent. We added iWildCam2020 from WILDS (Koh et al., 2021) in place of Camelyon17, using the predefined splits (covering distinct camera deployments).

For each dataset, we used the same model architecture as Rosenfeld & Garg (2023): ResNet or DenseNet, with a ResNet-50 pretrained on ImageNet for iWildCam. In most cases, models were pretrained on ImageNet. We trained or fine-tuned on source data using empirical risk minimization with data augmentation, excluding other training algorithms like FixMatch, BN-adapt, CDAN, and DANN for these experiments.

Critics were implemented as linear models consuming logits from the model under evaluation. We trained 30 randomly initialized critics in parallel and selected the best one. Following Rosenfeld & Garg (2023), the critics were trained for 100 epochs using the AdamW optimizer, with a learning rate of $3 \times 10^{-3}$ and weight decay of $5 \times 10^{-4}$. All loss functions involving softmax operations, including the standard cross-entropy loss and our proposed disagreement loss, are implemented in log-space to ensure numerical stability.

We attacked the target datasets using the approach detailed in Appendix F. We varied the fraction of attacked images $f$ in the target dataset over values $f \in \{0, 12.5, 25, 50\%\}$. In all cases, we ran the attack for 20 steps with a step size of $8/255$, constraining the magnitude of the perturbation for each image in $\ell_\infty$-norm to $4/255$. When updating the critic in each step, we used only 5 epochs, resulting in a total of 100 epochs of critic training over the course of the attack.

**Additional Results**    In Section 5.1.2, we compared surrogates based on their resulting estimates for the disagreement discrepancy, where larger estimates are superior. Figure 7 supplements these results with scatter plots comparing the complete error bound with the actual error on the attacked target data. Results are faceted by attack fraction $f$. We observe that the bound increasingly underestimates

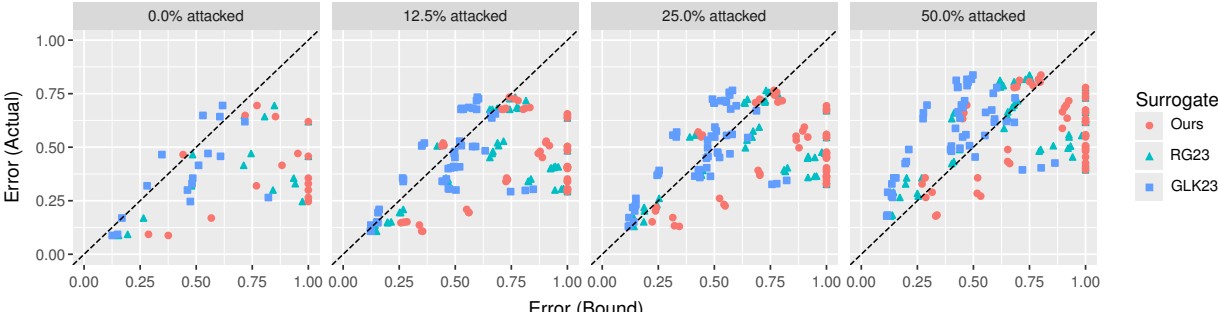

Figure 7: Comparison of error bounds versus actual error on attacked target data for our surrogate and that of Rosenfeld & Garg (2023). Each point represents a shift/model, with points above the dashed line indicating bound violations. Results are faceted by the fraction of attacked instances in the target data.

the actual error as $f$ increases, suggesting decreased robustness of the bound. Consistent with earlier results for disagreement discrepancy, our surrogate is least likely to underestimate the error, achieving the fewest violations for all values of $f$.

For completeness, we provide the results plotted in Figures 3 and 7 in tabular form in Table 2. Note that the attacks reported in the table are computed using our surrogate. We caution against comparing results at the level of source/target pairs, as we only run the attack and compute the bound once per pair, and there are several sources of randomness; instead, our analysis aggregates results across all pairs, providing a more robust statistical basis for our conclusions about the surrogates' performance.

### G.3 APPLICATION TO HARMFUL SHIFT DETECTION

To train the critic model using XGBoost, we implemented a custom objective function that defines the first-order gradient and second-order Hessian ($H$) of the loss with respect to the raw model logits. The total objective is a weighted sum over source samples $\mathcal{S}$ (where we minimize agreement loss) and target samples $\mathcal{T}$ (where we minimize disagreement loss). For a given input $x$, let $\mathbf{s} \in \mathbb{R}^K$ be the vector of logits output by the critic model, and let $y$ be the class predicted by the reference model. Let $\mathbf{p} = \sigma(\mathbf{s})$ be the predicted probabilities. We utilize a diagonal upper bound on the Hessian to ensure numerical stability and efficient tree splitting, as is standard in XGBoost implementations.

**Agreement Objective** For source samples, the critic minimizes the agreement loss $\ell_{\text{ce}}$ defined in (2). The gradient is the standard cross-entropy gradient. For the second-order term, we use the standard diagonal upper bound for all classes $k$:

$$\tilde{H}_k = 2p_k(1 - p_k).$$

**Disagreement Objective** For target samples, the critic minimizes a disagreement loss. The specific Hessian bound depends on the surrogate employed.

*Our Surrogate.* We minimize $\ell_{\text{dis}}^{\text{Ours}}$ defined in (9). We derived diagonal bounds on the Hessian to ensure convexity in the local neighborhood of the prediction:

$$\tilde{H}_k = \begin{cases} 2p_y(1 - p_y), & \text{if } k = y, \\ 2p_k p_y, & \text{if } k \neq y. \end{cases}$$

*GLK23 Surrogate.* The baseline minimizes $\ell_{\text{dis}}^{\text{GLK}}$ defined in (4). As this is functionally equivalent to a cross-entropy loss against a soft target distribution, we use the standard diagonal bound for all classes $k$:

$$\tilde{H}_k = 2p_k(1 - p_k).$$

| Source | Target | Attack (%) | Error (Actual) | Error (Bound) | | | Disagreement Discrepancy | | |
|---|---|---|---|---|---|---|---|---|---|
| | | | | GLK23 | Ours | RG23 | GLK23 | Ours | RG23 |
| cifar10 | cifar10_1v6 | 0 | 0.093 | 0.149 | 0.287 | 0.194 | 0.031 | 0.169 | 0.076 |
| | | 12.5 | 0.151 | 0.154 | 0.285 | 0.214 | 0.024 | 0.154 | 0.084 |
| | | 25 | 0.219 | 0.136 | 0.241 | 0.187 | 0.018 | 0.123 | 0.069 |
| | | 50 | 0.357 | 0.154 | 0.285 | 0.202 | 0.024 | 0.154 | 0.072 |
| | cifar10c_frost_level4 | 0 | 0.169 | 0.169 | 0.567 | 0.265 | 0.059 | 0.457 | 0.155 |
| | | 12.5 | 0.209 | 0.160 | 0.551 | 0.268 | 0.050 | 0.441 | 0.158 |
| | | 25 | 0.261 | 0.150 | 0.523 | 0.254 | 0.040 | 0.413 | 0.144 |
| | | 50 | 0.357 | 0.145 | 0.519 | 0.270 | 0.035 | 0.409 | 0.160 |
| | cifar10c_pixelate_level5 | 0 | 0.320 | 0.281 | 0.769 | 0.479 | 0.171 | 0.659 | 0.369 |
| | | 12.5 | 0.356 | 0.265 | 0.729 | 0.459 | 0.155 | 0.619 | 0.349 |
| | | 25 | 0.391 | 0.254 | 0.695 | 0.449 | 0.144 | 0.585 | 0.339 |
| | | 50 | 0.489 | 0.212 | 0.653 | 0.439 | 0.102 | 0.543 | 0.329 |
| | cifar10c_saturate_level5 | 0 | 0.088 | 0.126 | 0.376 | 0.154 | 0.016 | 0.266 | 0.044 |
| | | 12.5 | 0.136 | 0.124 | 0.340 | 0.142 | 0.014 | 0.230 | 0.032 |
| | | 25 | 0.171 | 0.119 | 0.315 | 0.140 | 0.009 | 0.205 | 0.030 |
| | | 50 | 0.290 | 0.118 | 0.316 | 0.143 | 0.008 | 0.206 | 0.033 |
| cifar100 | cifar100c_contrast_level4 | 0 | 0.330 | 0.481 | 1.000 | 0.942 | 0.200 | 0.969 | 0.661 |
| | | 12.5 | 0.408 | 0.519 | 1.000 | 0.959 | 0.236 | 0.971 | 0.677 |
| | | 25 | 0.469 | 0.570 | 1.000 | 0.998 | 0.289 | 0.984 | 0.717 |
| | | 50 | 0.616 | 0.682 | 1.000 | 1.000 | 0.399 | 0.987 | 0.775 |
| | cifar100c_motion_blur_level2 | 0 | 0.355 | 0.485 | 1.000 | 0.935 | 0.202 | 0.962 | 0.652 |
| | | 12.5 | 0.405 | 0.497 | 1.000 | 0.940 | 0.214 | 0.964 | 0.657 |
| | | 25 | 0.452 | 0.523 | 1.000 | 0.948 | 0.240 | 0.970 | 0.665 |
| | | 50 | 0.555 | 0.551 | 1.000 | 0.983 | 0.268 | 0.971 | 0.700 |
| | cifar100c_spatter_level2 | 0 | 0.300 | 0.461 | 1.000 | 0.842 | 0.178 | 0.913 | 0.560 |
| | | 12.5 | 0.352 | 0.460 | 1.000 | 0.837 | 0.177 | 0.911 | 0.554 |
| | | 25 | 0.404 | 0.466 | 1.000 | 0.826 | 0.183 | 0.914 | 0.543 |
| | | 50 | 0.507 | 0.483 | 1.000 | 0.856 | 0.200 | 0.922 | 0.573 |
| entity13_sub1 | entity13_sub2 | 0 | 0.465 | 0.347 | 0.442 | 0.484 | 0.093 | 0.187 | 0.230 |
| | | 12.5 | 0.519 | 0.362 | 0.446 | 0.441 | 0.108 | 0.192 | 0.187 |
| | | 25 | 0.571 | 0.332 | 0.428 | 0.424 | 0.078 | 0.174 | 0.169 |
| | | 50 | 0.697 | 0.293 | 0.470 | 0.449 | 0.039 | 0.216 | 0.195 |
| entity30_sub1 | entity30_sub2 | 0 | 0.648 | 0.530 | 0.718 | 0.715 | 0.146 | 0.334 | 0.330 |
| | | 12.5 | 0.681 | 0.528 | 0.726 | 0.687 | 0.144 | 0.342 | 0.302 |
| | | 25 | 0.724 | 0.492 | 0.702 | 0.648 | 0.108 | 0.318 | 0.264 |
| | | 50 | 0.812 | 0.423 | 0.705 | 0.619 | 0.039 | 0.321 | 0.235 |
| fmow_0212 | fmow_1315 | 0 | 0.415 | 0.511 | 0.883 | 0.712 | 0.068 | 0.440 | 0.269 |
| | | 12.5 | 0.480 | 0.497 | 0.878 | 0.668 | 0.054 | 0.436 | 0.225 |
| | | 25 | 0.552 | 0.487 | 0.864 | 0.664 | 0.044 | 0.422 | 0.221 |
| | | 50 | 0.686 | 0.454 | 0.907 | 0.672 | 0.011 | 0.464 | 0.230 |
| | fmow_1618 | 0 | 0.471 | 0.553 | 0.953 | 0.745 | 0.110 | 0.511 | 0.302 |
| | | 12.5 | 0.528 | 0.520 | 0.906 | 0.715 | 0.077 | 0.463 | 0.273 |
| | | 25 | 0.594 | 0.502 | 0.901 | 0.700 | 0.060 | 0.459 | 0.257 |
| | | 50 | 0.717 | 0.461 | 0.925 | 0.697 | 0.018 | 0.483 | 0.254 |
| iwildcam2020 | iwildcam2020 | 0 | 0.265 | 0.822 | 1.000 | 1.000 | 0.493 | 0.790 | 0.726 |
| | | 12.5 | 0.304 | 0.848 | 1.000 | 1.000 | 0.518 | 0.763 | 0.738 |
| | | 25 | 0.345 | 0.823 | 1.000 | 1.000 | 0.493 | 0.765 | 0.723 |
| | | 50 | 0.426 | 0.863 | 1.000 | 1.000 | 0.534 | 0.777 | 0.727 |
| living17_sub1 | living17_sub2 | 0 | 0.695 | 0.618 | 0.771 | 0.849 | 0.159 | 0.312 | 0.390 |
| | | 12.5 | 0.734 | 0.598 | 0.741 | 0.768 | 0.139 | 0.282 | 0.309 |
| | | 25 | 0.756 | 0.567 | 0.778 | 0.743 | 0.108 | 0.319 | 0.284 |
| | | 50 | 0.837 | 0.498 | 0.801 | 0.750 | 0.039 | 0.342 | 0.291 |
| nonliving26_sub1 | nonliving26_sub2 | 0 | 0.643 | 0.606 | 0.854 | 0.805 | 0.262 | 0.510 | 0.461 |
| | | 12.5 | 0.685 | 0.559 | 0.833 | 0.774 | 0.216 | 0.489 | 0.430 |
| | | 25 | 0.717 | 0.529 | 0.803 | 0.728 | 0.186 | 0.459 | 0.384 |
| | | 50 | 0.804 | 0.476 | 0.797 | 0.696 | 0.133 | 0.453 | 0.352 |
| officehome_product | officehome_art | 0 | 0.457 | 0.609 | 1.000 | 1.000 | 0.390 | 0.973 | 0.910 |
| | | 12.5 | 0.483 | 0.633 | 1.000 | 1.000 | 0.414 | 0.966 | 0.910 |
| | | 25 | 0.545 | 0.649 | 1.000 | 1.000 | 0.430 | 0.959 | 0.905 |
| | | 50 | 0.621 | 0.647 | 1.000 | 1.000 | 0.428 | 0.957 | 0.912 |
| | officehome_clipart | 0 | 0.619 | 0.717 | 1.000 | 1.000 | 0.491 | 0.955 | 0.921 |
| | | 12.5 | 0.636 | 0.663 | 1.000 | 1.000 | 0.448 | 0.957 | 0.896 |
| | | 25 | 0.670 | 0.688 | 1.000 | 1.000 | 0.462 | 0.941 | 0.892 |
| | | 50 | 0.730 | 0.685 | 1.000 | 1.000 | 0.471 | 0.957 | 0.903 |
| | officehome_real | 0 | 0.246 | 0.474 | 1.000 | 0.974 | 0.259 | 0.863 | 0.759 |
| | | 12.5 | 0.300 | 0.503 | 1.000 | 0.944 | 0.277 | 0.867 | 0.718 |
| | | 25 | 0.364 | 0.483 | 1.000 | 0.947 | 0.268 | 0.860 | 0.732 |
| | | 50 | 0.454 | 0.508 | 1.000 | 0.942 | 0.293 | 0.869 | 0.727 |

Table 2: Comparison of target error bounds and estimated disagreement discrepancies across different surrogates for various source/target data pairs. For attack rates greater than 0%, the target datasets are adversarially perturbed using our proposed disagreement loss and surrogate for disagreement discrepancy.

## H    USE OF LARGE LANGUAGE MODELS

We used large language models (LLMs) as writing assistance tools to help synthesize and polish text based on author-provided content, including drafts, bullet points, and technical explanations. LLMs were also employed to review mathematical proofs for potential errors. All substantive content, theoretical contributions, experimental design, and scientific conclusions are the original work of the authors. The LLMs did not contribute to the conceptual development of the research or the generation of novel ideas, and their usage does not constitute authorship-level contribution.

