# OpenReview forum: "On the Bayes Inconsistency of Disagreement Discrepancy Surrogates"
_ICLR.cc/2026/Conference — ICLR 2026 Poster_

### Official Review · Reviewer_RFmD · 2025-10-30

**Soundness:** 4
**Presentation:** 4
**Contribution:** 3
**Rating:** 8
**Confidence:** 3

**Summary:**

The paper considers the problem of measuring maximal discrepancy in covariate shifts in the context of bounding generalization errors, noting that the theory is based on 0-1 losses, but surrogate losses used in practice, while convex, are not consistent. That is, the optimizer of the surrogate loss does not converge to the solution of 0-1 loss due to the choice of the disagreement loss. A consistent disagreement loss is proven and shown to perform better in estimating the actual discrepancy.

**Strengths:**

The paper is clear and address a perennial problem of understanding how covariate shift can affect classifiers. Overall, the paper is well written.

**Weaknesses:**

There are a few points that could be made more clear (questions below).

Additionally, is it possible to use the consistent form as a bound on error to create distributional robustness? I understand the adversarial attack is a step in this direction, but commenting on how this connects to making better unsupervised domain adaption algorithms would be interesting.

**Questions:**

What is the purpose of minimum in (1) and if it is needed what is it taken with respect to what?

Line 95 The limitations of theory is noted but could be made clearer on why 'practical model classes' are note covered by $\mathcal{H}$-consistency. "While H-consistency provides valuable insights, its limited applicability to practical model classes" It is not until line 493 that deep neural networks are mentioned.

Line 111 Could be more clear by defining what exactly "this framework" is. I understand it to be the description at line 104–105.

Line 132 The description of a probability "intersected with $\mathcal{X}'$ is strange and imprecise. Why not say it is the conditional distribution for points in the subset $\mathcal{X}'$.

Line 399 The critic was using the frozen weights. In cases where the reference model like ERM is not aware of the target data, couldn't a better critic be built on top of the backbone optimized in a different unsupervised domain adaption method? I.e. couldn't $f$ be used as a critic for $h$ even though $f$ uses $h'$?

Line 1616 the $\ell_\infty$ should be on both target training and target testing as both are optimized in the two steps of the optimization.

Line 1621 Couldn't the adjoint/ implicit function method  be used here?

---

> ### Author Response · Authors · 2025-11-19
> **Initial response**
>
> We thank the reviewer for their careful reading of our work and their many insightful comments and suggestions.
>
> > Additionally, is it possible to use the consistent form as a bound on error to create distributional robustness?
>
> This is an excellent point: you are right to connect our work to robust training. Methods like D-BAT (Pagliardini et al., 2023), which we cite, use disagreement-based objectives for this purpose. However, their training objective relies on a surrogate that lacks a formal proof of Bayes consistency. This introduces a potential risk that the optimization may not faithfully track the true disagreement objective. Our work addresses this by providing a provably consistent surrogate, which can be used as a drop-in replacement. We have added a sentence to our discussion of Pagliardini et al. (2023) in §2.2, clarifying this point.
>
> > What is the purpose of minimum in (1) and if it is needed what is it taken with respect to what?
>
> Thank you for catching this ambiguity. The $\min$ is a tie-breaking mechanism. Since $\arg \max$ can return a set of indices if multiple logits share the same maximum value, we take the $\min$ (i.e., the smallest index) from that set to ensure a deterministic output. We’ve added a sentence below Equation (1)  to clarify this point.
>
> > Line 95 The limitations of theory is noted but could be made clearer on why 'practical model classes' are note covered by $\mathcal{H}$-consistency.
>
> We have updated the text in §2.1 to clarify this point. Our statement reflects a well-understood challenge in learning theory. $\mathcal{H}$-consistency analysis provides refined guarantees, but the proofs typically require strong structural assumptions about the hypothesis class $\mathcal{H}$ (e.g., related to convexity or its complexity measures). While this analysis has been successfully applied to classes like linear models or 1-layer neural networks, these assumptions are generally not met by the highly complex, non-convex function classes represented by the deep neural networks used in modern practice. To our knowledge, no successful $\mathcal{H}$- -consistency analysis currently exists for these architectures.
>
> > Line 111 Could be more clear by defining what exactly "this framework" is...
>
> You are correct. “This framework” refers specifically to the approach of maximizing the disagreement discrepancy between a trainable critic and a fixed reference model. We have revised the sentence for clarity.
>
> > Line 132 The description of a probability "intersected with $\mathcal{X}’$ is strange and imprecise.
>
> Thanks for pointing this out. To avoid ambiguity we have added a footnote: “Formally, for any event $A$, we define $S|_{\mathcal{X}'}(A) = S(A \cap \mathcal{X}') / S(\mathcal{X}')$ assuming $S(\mathcal{X}') > 0$”.
>
> > Line 399 ... In cases where the reference model like ERM is not aware of the target data, couldn't a better critic be built on top of the backbone optimized in a different unsupervised domain adaption method?
>
> This is an excellent point on critic flexibility. Rosenfeld & Garg’s framework is indeed flexible regarding the choice of critic hypothesis class $\mathcal{H}$, with the key theoretical requirement being its expressiveness.
>
> Our primary goal, however, was to perform a controlled experiment that _isolates the impact of the surrogate loss_. For this reason, we deliberately replicated the experimental setup of Rosenfeld & Garg (2023), including their choice of critic architecture. This ensures a fair, apples-to-apples comparison, demonstrating that simply swapping an inconsistent surrogate for our consistent one yields significant improvements.
>
> That said, we’d like to highlight that our evaluation was conducted on a diverse set of reference models. As detailed in Appendix G.1, these models were trained not just with ERM, but also with domain adaptation methods: DANN, CDAN, FixMatch and BN-adapt. The consistent performance improvement of our surrogate across these varied settings speaks to its general utility.
>
> > Line 1616 the $\ell_\infty$ should be on both target training and target testing as both are optimized in the two steps of the optimization.
>
> You are correct. Thank you for spotting this oversight.
>
> > Line 1621 Couldn't the adjoint/ implicit function method be used here?
>
> You are correct that the implicit function theorem provides a way to differentiate through the solution of an optimization problem like the one used to find the optimal critic $f^\star$.
>
> However, our understanding is that current frameworks (e.g., JAXopt, TorchOpt) require the parameters to fit in memory. Our attack perturbs a large fraction (up to 50%) of the target dataset, which is too large to be stored in GPU memory simultaneously. For this practical engineering reason, we opted for a two-step iterative procedure that could be performed in batches, which scales to the large datasets used in our experiments.

---

> > ### Comment · Reviewer_RFmD · 2025-11-26
> >
> > Thanks for the response, I'll be maintaining my score.

---

### Official Review · Reviewer_cJ7E · 2025-10-31

**Soundness:** 3
**Presentation:** 3
**Contribution:** 3
**Rating:** 4
**Confidence:** 3

**Summary:**

The paper studies the use of surrogate losses to maximize disagreement discrepancy - a quantity used for error bounds under covariate shift, shift detection, and robust training. The authors claim to prove that commonly used disagreement surrogates are not Bayes consistent for multi-class problems and develop upper/lower bounds on the optimality gap. Guided by this analysis, they propose a new disagreement loss, $ ℓ^{ours}_{​dis}(x,y,s)=−log(1−σ(s)_y​) $
which, when paired with cross-entropy agreement loss, yields a surrogate provably Bayes-consistent. Empirical results on a variety of vision benchmarks (WILDS, BREEDS, DomainNet) and adversarially perturbed target data show improved estimation of the disagreement discrepancy and better calibration of the associated error bounds versus prior surrogates.

**Strengths:**

The paper tackles a fundamental open question in the theory of domain adaptation and covariate shift: whether the surrogate losses used for estimating disagreement discrepancy are theoretically sound.
Few prior works examined the consistency properties of such surrogates; most focused on empirical performance.
The authors provide what appears to be the first demonstration that the popular disagreement surrogates of Rosenfeld & Garg (2023) and Ginsberg et al. (2023) are Bayes-inconsistent in the multi-class (K > 2) case.
This negative result is conceptually important because it challenges implicit assumptions underlying a growing body of empirical research.
The proposed alternative surrogate loss, proven (under stated assumptions) to be Bayes-consistent, gives the work a constructive dimension - turning a theoretical diagnosis into a practical remedy.
While the proofs appear well-structured and follow logically from standard surrogate-consistency frameworks, a full validation would require a detailed check of each derivation and of how the assumptions are applied. The arguments look rigorous and well-motivated, but independent verification is still needed to certify formal correctness.

**Weaknesses:**

The main theoretical guarantee - Bayes consistency - is asymptotic and does not specify convergence rates or finite-sample behavior.
This makes it unclear how the surrogate behaves with limited data or under model misspecification. Without sample-complexity or generalization bounds, practitioners lack quantitative guidance on when the theoretical improvement translates into measurable gains.

The analysis assumes covariate shift only, i.e., $P(Y \textbar X) $ remains constant across domains. In real-world distribution shifts, label shift, concept drift, or conditional shift frequently occur - cases where the proposed theory may not apply or may even break down. No empirical evidence is provided for robustness beyond this assumption.

The paper largely dismisses H-consistency (consistency under restricted hypothesis classes) as "limited in practice" without detailed justification or experiments comparing H-consistent and Bayes-consistent outcomes. Since DNNs are not universal function approximators in practice, H-consistency might be more relevant than Bayes consistency.

The results depend on the structure of discrete label sets and the softmax parameterization. It is unclear whether the framework extends to regression, structured prediction, or representation-learning contexts where disagreement discrepancy is also used.

While the experimental results are broad in scope, they are not statistically rigorous. Most results are reported as single runs, without standard deviations, making it difficult to assess robustness or reproducibility. The paper would benefit from multi-seed evaluations and statistical significance tests.

Hyperparameter sensitivity is not explored: the method introduces parameter $\alpha$ whose effect on stability and performance remains unclear.

There is no discussion of computational overhead (e.g., runtime, convergence rate) introduced by the new surrogate.
The theoretical contribution is strong, but the empirical evidence supporting its practical relevance remains somewhat limited in depth and statistical reliability.

The paper compares primarily with Rosenfeld & Garg (2023) and Ginsberg et al. (2023), which are relevant but now slightly dated.
More recent 2024–2025 methods tackle disagreement-based discrepancy estimation and should be included for completeness.
Without these baselines, it is difficult to assess whether the proposed surrogate remains competitive with the current state of the art in practical terms.

The lack of up-to-date comparisons weakens the empirical claim of superiority, even though the theoretical advance remains valid.
Overall, the paper’s main weaknesses lie not in its conceptual or theoretical core, which is solid, but in its empirical completeness and clarity of assumptions. Statistical validation is thin, comparisons are slightly outdated, and the theoretical applicability to finite neural critics could be more clearly discussed. These issues do not undermine the core contribution but do limit the paper’s perceived maturity and readiness for publication at the very top tier.

**Questions:**

1) How general is the proposed framework for difference-of-risk objectives? Could it apply to other discrepancy-based metrics (e.g., fairness gaps, calibration errors), or is it specific to disagreement discrepancy?
2) Theorems 4 and 6 provide lower and upper bounds on optimality gaps. Are these bounds tight in practice, or are they mostly asymptotic constructs with little finite-sample relevance?
3) How sensitive are your results to the choice of the critic’s hypothesis class H? Have you considered the notion of H-consistency or provided any analysis for finite-capacity critics?
4) Does the new surrogate require calibration of $\alpha$ to maintain consistency, or is it theoretically invariant to this hyperparameter?
5) What guided the decision to use a frozen feature extractor and a linear critic head? Would your results change with a trainable or deeper critic?
6) Can you provide mean ± std results over multiple random seeds for key tables and figures?
7) Have you compared against any of the newer disagreement-based methods?
8) Would it be possible to include a visual example or schematic showing how your surrogate modifies the disagreement landscape compared to Rosenfeld & Garg’s surrogate?
9) Do you expect your theoretical framework to extend naturally to other discrepancy measures?
10) Could your approach provide practical benefits in domains beyond covariate shift?

---

> ### Author Response · Authors · 2025-11-14
> **Clarification on 2024/2025 methods**
>
> Thank you for your detailed feedback on our submission. We appreciate the time you've taken to review our work and are currently drafting a comprehensive response.
>
> To ensure we can address your comments as thoroughly as possible, we were hoping you could provide clarification on one point. Specifically, you mention that our comparisons are with "slightly dated" methods and that "More recent 2024–2025 methods tackle disagreement-based discrepancy estimation and should be included for completeness."
>
> We'd appreciate it if you could point us to specific papers or methods from 2024–2025 you have in mind. Any examples you could provide would be helpful for us to situate our work within the current state of the art.
>
> Best regards,
> The Authors

---

> ### Author Response · Authors · 2025-11-19
> **Initial response**
>
> Thank you for this exceptionally thorough review. We address the main themes from your "Weaknesses" section first, then your specific questions.
>
> ### Weaknesses
>
> 1. **Focus on Bayes consistency.** You are correct that our result is asymptotic. We have revised the paper (see §1, §2.1, and below Theorem 6) to clarify the role of our work within the broader landscape of learning theory. A full finite-sample guarantee is formed by bounding two components: the _estimation error_ (from finite data) and the _calibration error_ (from using a surrogate). Our Theorem 6 provides the latter. The primary bottleneck to a full guarantee for deep learning is bounding the estimation error, which is an open challenge. By providing the first sound bound on the calibration error for disagreement discrepancy, we lay the essential groundwork for future analyses.
>
> 2. **Statistical rigor.** We respectfully disagree that our validation lacks rigor; rather, our rigor comes from its breadth. Evaluating across 130 distinct model/shift scenarios (in §5.1) provides a more robust and generalizable test of utility than running multiple seeds on a handful of benchmarks. While a single run can be noisy, the strong aggregate trend across these diverse scenarios provides a powerful high-level signal. To make this explicit, we have supplemented our aggregate rank results from Figs. 1 and 3 with a formal significance test where we find p-values < 3e-4 in all cases.
>
> 3. **Covariate shift setting.** This is the standard assumption in the line of work we build upon (Rosenfeld & Garg, 2023; Ginsberg et al., 2023). Our goal was to rigorously analyze and improve a core component _within_ this established framework. Extending the disagreement discrepancy framework to handle more complex shifts (e.g., label or concept shift) is an important but orthogonal research direction.
>
> 4. **$\alpha$ hyperparameter.** We clarify that $\alpha$ is not a hyperparameter we introduce, but a parameter of the disagreement discrepancy objective itself, present in prior work (Ginsberg et al., 2023; Pagliardini et al., 2023). Our Theorem 6 holds for all $\alpha > 0$. The theory behind the error bound in §5 _requires_ $\alpha = 1$, so the value is a matter of correctness, not tuning.
>
> ### Questions
>
> > How general is the proposed framework for difference-of-risk objectives?
>
> Yes, we are confident that the core of our theoretical framework is broadly applicable to other objectives that can be formulated as a difference-of-risks.
>
> The key insight of our approach is the reformulation of any difference-of-risks objective as a single, standard risk objective using what we term "pseudo-losses" (see §4.2). Once the problem is in this standard form, one can:
>
> - Proving inconsistency using our general lower-bounding technique (Appendix A).
>
> - Proving consistency following the structure of our excess error bound proof (Theorem 6). The details would depend on the functional form of the new losses.
>
>  Therefore, while we focus on the important and previously unsolved problem of disagreement discrepancy, the analytical framework we developed is designed to be more general.
>
> > Are these bounds [Theorems 4 and 6] tight in practice, or are they mostly asymptotic constructs...?
>
> These bounds are indeed primarily asymptotic analytical tools, but they have a very specific and crucial role. Their purpose is to formally bound the calibration error between the surrogate and target objectives. While they are not full finite-sample guarantees on their own, they are the essential theoretical link required to create them. As we now clarify in the paper (below Theorem 6), a full guarantee could be formed by combining our excess error bound with a second bound on the statistical error (the gap between empirical and true risk).
>
> > What guided the decision to use a frozen feature extractor and a linear critic head?
>
> Our guiding principle was to perform a _controlled comparison_. To isolate the impact of the surrogate loss, we deliberately replicated the setup of Rosenfeld & Garg (2023), who made this design choice. This included using both logit-based and feature-based linear critics (see Figure 5). Innovating on critic architectures is an interesting but orthogonal research direction.
>
> > Have you compared against any of the newer disagreement-based methods?
>
> Ensuring our comparisons are with the current state-of-the-art is a priority. We posted a clarification request at the start of the response period regarding the specific newer methods you have in mind. We look forward to your guidance on this point.
>
> > Would it be possible to include a visual example or schematic...?
>
> This is an interesting idea. However, the loss landscape is a function of the high-dimensional logit vector $\mathbf{s}$, making it difficult to visualize faithfully without arbitrary and potentially misleading low-dimensional projections.

---

> > ### Comment · Reviewer_cJ7E · 2025-11-27
> > **Response to Author.**
> >
> > I highly appreciate the authors’ effort in preparing the rebuttal.
> > I must admit that I am not an expert in this topic. However, this paper was assigned to me. The questions I raised were mainly for clarification and you have addressed all of them. Therefore, I consider raising my score.

---

### Official Review · Reviewer_H4wG · 2025-11-01

**Soundness:** 3
**Presentation:** 2
**Contribution:** 2
**Rating:** 6
**Confidence:** 4

**Summary:**

The paper studies disagreement discrepancy, an approach that has been recently popular for distribution shifts problem related to error estimation, testing, and robustness. Earlier works propose to maximise the disagreement discrepancy by employing some surrogate loss function, however the main finding of this paper is that the employed surrogate may not be Bayes consistent, which is necessary. The paper then  proposes a Bayes consistent surrogate for the problem.

**Strengths:**

1. I think paper makes clear theoretical contribution, which is also important. I've seen the works on disagreement discrepancy before as applied for testing, and have found the application to be hacky. The current contribution studies the soundness of it, and indeed, the finding that prior approaches are not sound is original and of significance.
2. The paper is also thorough (I like the appendix, although I didn't verify everything too closely), but the decomposition of the risk using density ratios, and the resulting Bayes inconsistency analysis seems correct to me. In that sense, the proposed surrogate is also sound (wonder why do the authors think, that was not already employed in prior works as a disagreement loss, as it seems super intuitive).

**Weaknesses:**

1. Overall, I do like the paper from the theoretical contribution perspective. However, I'm not sure I followed the experimental section that well. Could authors clarify what is max disagreement discrepancy (in Figure 1) and how it is defined and estimated? I also feel writing wise, the paper assumes too much background on prior work.
2. Furthermore, I think while the quality of surrogate is the main focus of the paper, the paper does not connect to downstream applications. I'd assume in testing works, the proposed surrogate could help aid better testing power.
3. Additionally, I feel like the new disagreement loss can be unstable as $\sigma(s)_{y}$ approaches 1, which it can as neural networks can get overconfident. Has that been considered before? I assume that could or could not happen, and in fact, maybe the proposed disagreement could even lead to better calibrated probability estimates? I'd be curious to learn some perspectives on this.

Summary: I think the paper makes clean theoretical contribution, that is useful. But from the practical side, I can have a better assessment if the above questions are addressed in the rebuttal.

**Questions:**

In Equation 1, what is the $\min$ over?

---

> ### Author Response · Authors · 2025-11-19
> **Initial response**
>
> We thank the reviewer for their thoughtful feedback. We are encouraged that the reviewer found our theoretical contributions to be significant.
>
> > Could authors clarify what is max disagreement discrepancy (in Figure 1) and how it is defined and estimated?
>
> We agree that this deserves clarification. The "max disagreement discrepancy" for a given scenario is the maximum value achieved among all the surrogates we tested. Since the true maximum over the critic hypothesis class is intractable, this serves as a practical benchmark to directly compare the relative performance of the methods. A surrogate performs better when its estimate is closer to this benchmark.
>
> We have added clarifying sentences in §5.1 and §5.2 to make this definition clear within the text.
>
> > ...the paper assumes too much background on prior work.
>
> Thank you for raising this. Ensuring the paper is self-contained is a high priority.
>
> Based on feedback from all reviewers, we have already taken steps in our revision to improve accessibility:
>
> - We added a new paragraph to the Introduction to better motivate the theoretical landscape of consistency.
>
> - We expanded the related work (§2.1) to provide a more pedagogical explanation of consistency guarantees.
>
> - We revised the opening of §5 to frame our experiments as a downstream application case study.
>
> We hope these revisions address your concerns. We welcome any further suggestions you may have, noting that we must balance them with the page space required for new experiments based on Ginsberg et al. (2023) (see below).
>
> > The paper does not connect to downstream applications.
>
> As detailed in our Introduction (lines 38-41), our work is directly motivated by the need to improve a suite of downstream applications. These include:
>
> - _Bounding model error on unlabeled target data_: for assessing performance and making deployment decisions when new labels are costly or unavailable.
>
> - _Statistical testing for harmful shifts_: for monitoring deployed models in production to detect silent failures before they cause harm.
>
> - _Training robust model ensembles_: to improve generalization, mitigate shortcut learning, and enhance transferability to new domains.
>
> Our empirical work in §5 is dedicated to the first and most direct of these applications: error bounding. We chose this application for our primary evaluation because the _validity_ and _reliability_ of the error bound depend strictly on the accurate maximization of the disagreement discrepancy, making it the most rigorous test of a surrogate's quality.
>
> To further strengthen this connection based on your feedback, we have taken two actions in our revision:
>
> 1. As noted in our response to your point on "testing power," we are adding a new evaluation on the statistical testing application (following Ginsberg et al., 2023).
>
> 2. We have revised the introduction of §5 to more explicitly frame our experiments as a downstream application case study. We now highlight how the reliability and validity of the error bound depend directly on the quality of the surrogate, clarifying the practical stakes of our theoretical contribution.
>
> > I'd assume in testing works, the proposed surrogate could help aid better testing power.
>
> We agree that a more accurate estimate of disagreement discrepancy, which our surrogate provides, should translate into a more powerful statistical test in the framework of Ginsberg et al. (2023). To that end, we have begun running experiments using Ginsberg et al.’s code and our proposed surrogate. We will post another revision of our paper with these new experimental results before the end of the paper discussion phase.
>
> > I feel like the new disagreement loss can be unstable as $\sigma(s)_y$ approaches 1...
>
> The design of our disagreement loss, $- \log (1 -\sigma(\mathbf{s})_y)$, is intentionally symmetric to the standard cross-entropy agreement loss, $-\log(\sigma(\mathbf{s})_y)$. It can be interpreted as the cross-entropy of a binary problem where the "positive" class is $y$ and the "negative" class is the collection of all other classes. Therefore, its numerical stability is equivalent to that of standard cross-entropy. You are right to consider the potential for instability with extreme probabilities. This is a well-known issue for all cross-entropy-style losses. For our implementation, we use the standard technique of performing all calculations in log-space for numerical stability. We have added a brief note to Appendices G.1 and G.2 confirming this implementation detail.
>
> > In Equation 1, what is the $\min$ over?
>
> Thank you for catching this ambiguity. The $\min$ is a tie-breaking mechanism. Since $\arg \max$ can return a set of indices if multiple logits share the same maximum value, we take the $\min$ (i.e., the smallest index) from that set to ensure a deterministic output. We’ve added a sentence below Equation (1) to clarify this point.

---

> > ### Author Response · Authors · 2025-11-29
> > **A second revision with new experiments**
> >
> > To close the loop on our previous response, we have now completed the promised experiments on testing for harmful shift and have uploaded a second revision of our manuscript.
> >
> > For clarity, here is a summary of the key updates:
> >
> > - We added a new Section 5.2, "Application: Detecting Harmful Covariate Shift," dedicated to this new experiment, which was inspired by your feedback.
> > - Following the protocol of Ginsberg et al. (2023), our results (Figure 4, Table 1) show that our consistent surrogate leads to a statistically significant improvement in statistical power, demonstrating a clear practical benefit on a second downstream task.
> > - We added a new Appendix G.3 with implementation details and updated the introduction and conclusion to reflect this new contribution.
> >
> > We are aware of the procedural changes to the review process and understand that further discussion or score updates are not possible. Nevertheless, we wanted to demonstrate that we took your feedback seriously. We believe these revisions comprehensively address the concerns you raised and have substantially strengthened the paper. Under normal circumstances, we are confident that these improvements would merit a re-evaluation of the reviewer's score.

---

### Author Response · Authors · 2025-11-29

Given the procedural changes to the review process, we would like to provide a summary of our revisions to assist with the final decision.

## Summary of Major Revisions

Our revisions were guided by reviewer feedback and focused on three main themes:
- _Strengthening connection to downstream applications._ We added a major new experimental section (Section 5.2) evaluating our surrogate on harmful shift detection (following Ginsberg et al., 2023). The results show a statistically significant improvement in statistical power, demonstrating a clear practical benefit on a second important application.
- _Theoretical positioning._ We expanded our discussion on learning theory in the introduction, related work (Section 2.1), and theory (below Theorem 6) to more clearly articulate the hierarchy of consistency guarantees and the role of our work.
- _Improving presentation and clarity._ We incorporated numerous smaller revisions to improve accessibility, including clarifying key definitions, adding implementation details to the appendix, and refining the framing of our experiments.

## Core Strengths

Even before our revisions, the reviewers highlighted several core strengths of our work, noting that it makes a "clear theoretical contribution" (H4wG) and is "well written" (RFmD). They also highlighted that the paper provides "the first demonstration that the popular disagreement surrogates... are Bayes-inconsistent" and successfully "[challenges] implicit assumptions underlying a growing body of empirical research" while "turning a theoretical diagnosis into a practical remedy" (cJ7E).

## Addressing Reviewer Concerns
We are confident our revisions have fully addressed the reviewers' initial concerns.

- **Reviewer cJ7E** raised several detailed questions, primarily concerning the asymptotic nature of our theory and the statistical rigor of our experiments. We were encouraged that, after our comprehensive revisions and point-by-point rebuttal, the reviewer confirmed we **"have addressed all of them"** and stated their intention to raise their score.
- **Reviewer H4wG** primary concern was the paper's connection to downstream applications. Our most substantial revision directly addresses this by adding a **new experimental section (5.2)** inspired by their feedback. Due to the premature conclusion of the discussion period, the reviewer did not have an opportunity to respond to this major update. However, given that this revision comprehensively resolves their main reservation, we are confident it would have been received positively.

The final manuscript now not only provides the first provably Bayes-consistent surrogate for disagreement discrepancy - resolving a foundational flaw in prior work - but also demonstrates its practical benefits across two critical applications, solidifying its contribution to the field.

---

### Meta-Review · Area_Chair_CaTn · 2026-01-11

**Summary:**

A few main concerns:

* While the quality of surrogate is the main focus of the paper, the paper does not connect to downstream applications. I'd assume in testing works, the proposed surrogate could help aid better testing power.
* the new disagreement loss can be unstable, which it can as neural networks can get overconfident. Has that been considered before?
* The main theoretical guarantee - Bayes consistency - is asymptotic and does not specify convergence rates or finite-sample behavior. This makes it unclear how the surrogate behaves with limited data or under model misspecification.
* While the experimental results are broad in scope, they are not statistically rigorous.

**Reviewer Concerns:**

I think most of their concerns were answered

**Reviewer Scores:**

the reviewer scores would have changed to 6,6,8 (avg 6.66), could go even higher.

---

### Decision · Program_Chairs · 2026-01-26

Accept (Poster)